# The ER–Golgi intermediate compartment is a key membrane source for the LC3 lipidation step of autophagosome biogenesis

**Liang Ge, David Melville, Min Zhang, Randy Schekman\***

Department of Molecular and Cell Biology, Howard Hughes Medical Institute, University of California, Berkeley, Berkeley, United States

**Abstract** Autophagy is a catabolic process for bulk degradation of cytosolic materials mediated by double-membraned autophagosomes. The membrane determinant to initiate the formation of autophagosomes remains elusive. Here, we establish a cell-free assay based on LC3 lipidation to define the organelle membrane supporting early autophagosome formation. In vitro LC3 lipidation requires energy and is subject to regulation by the pathways modulating autophagy in vivo. We developed a systematic membrane isolation scheme to identify the endoplasmic reticulum–Golgi intermediate compartment (ERGIC) as a primary membrane source both necessary and sufficient to trigger LC3 lipidation in vitro. Functional studies demonstrate that the ERGIC is required for autophagosome biogenesis in vivo. Moreover, we find that the ERGIC acts by recruiting the early autophagosome marker ATG14, a critical step for the generation of preautophagosomal membranes.

## Introduction

Autophagy is a conserved catabolic process underlying the self-digestion of cytoplasmic components through the formation of double-membraned vesicles termed autophagosomes. One basic role of autophagy is to turn over damaged proteins and organelles to maintain cellular homeostasis. Autophagy also allows cells to cope with stresses such as starvation, hypoxia, and pathogen infection (*Mizushima et al., 2008*; *Burman and Ktistakis, 2010*; *Levine, 2005*; *Yang and Klionsky, 2010*; *Weidberg et al., 2011*).

In the process of starvation-induced autophagy, several upstream signals are triggered, including inhibition of the mechanistic target of rapamycin (MTOR), and activation of the Jun N-terminal kinase (JNK) and AMP kinase (AMPK) (*Noda and Ohsumi, 1998*; *Wei et al., 2008*; *Kim et al., 2011*; *Rubinsztein et al., 2012*; *Kim et al., 2013*). These signals are conveyed to activate the serine/threonine-protein kinase complex containing the Atg1 homologs ULK1/2, ATG13, FIP200 (RB1CC1) and ATG101 (C12orf44) (*Mizushima, 2010*). Together with upstream signals, this complex promotes the formation and membrane docking of the class III phosphatidylinositol 3 (PtdIns3)-kinase (PI3K) complex consisting of ATG14 (ATG14L/Barkor), the Atg6 homologue BECN1 (Beclin1), VPS34 (PIK3C3) and VPS15 (p150) for phosphatidylinositol 3-phosphate (PI3P) production (*Obara and Ohsumi, 2011*). Subsequently, DFCP1 (ZFYVE1), an endoplasmic reticulum (ER)-associated PI3P binding protein, is recruited to the site of newly-generated PI3P to form omegasomes (*Axe et al., 2008*). This is followed by two ubiquitin-like conjugation systems to produce the ATG12–ATG5 conjugate and phosphatidylethanolamine (PE)-lipidated ATG8/LC3, which initiates the formation of a preautophagosomal organelle termed the phagophore or isolation membrane (*Mizushima et al., 1998a*, *1998b*; *Ichimura et al., 2000*; *Geng and Klionsky, 2008*). The membrane then expands and engulfs cytoplasmic components. Finally, the crescent-shaped tubular membrane seals to form a double-membraned autophagosome with cytoplasmic

\*For correspondence: schekman@berkeley.edu

**eLife digest** Cells continually adapt their behavior to accommodate changes in their environment. For example, when nutrients are abundant, cells can grow or proliferate; in times of scarcity, however, they must conserve resources for essential tasks. In particular, during periods of starvation, cells can cannibalize themselves in a process called autophagy, which literally means 'self-eating'. Structures called autophagosomes engulf bits of cytoplasm and carry the contents to the digestive compartment of the cell, the lysosome, to be broken down into their constituent parts. This can include the degradation of proteins into amino acids, which can then be recycled into other proteins needed by the cell.

In cells, proteins are shipped to their destinations—which can be the plasma membrane or a specific organelle within the cell—via a delivery system known as the secretory pathway. This pathway begins in the endoplasmic reticulum (ER), where many of these proteins are made. From the ER, the proteins move to a compartment called the Golgi apparatus, which then sends them to their destinations, or to the lysosome to be broken down. Between the ER and Golgi they pass through a structure called the ER–Golgi intermediate compartment (ERGIC).

Although the signaling pathways that initiate autophagy are known, less is understood about the actual formation of the autophagosomes. Now, Ge et al. have developed an in vitro system to study their formation, and gone on to identify a membrane that is both necessary and sufficient to create these structures.

Previous studies have implicated a variety of membranes—including the plasma membrane and the membranes belonging to the ER, the Golgi apparatus, the lysosome and various other organelles—in the formation of autophagosomes. To identify which of these membranes might be involved, Ge et al. focused on a protein called LC3 that is a key marker for the formation of the autophagosome. This protein is recruited to the growing autophagosome by a lipid, so discovering which membranes can add a lipid to LC3 should shed light on the assembly process.

By separating the full range of organelles in a cell lysate into fractions (a process called fractionation), Ge et al. found that the ERGIC was the most active membrane to attach lipid to LC3. Additionally, the lipid was only added when signaling pathways that stimulate autophagy—such as the PI3K pathway—were activated. Together, these results provide insight into the mechanism of autophagosome formation, and the structures in the cell that participate in this process.

components enclosed within the inner membrane. Fusion of the autophagosome with the lysosome leads to the breakdown of the inner membrane together with the trapped cytosolic material (*Mizushima et al., 1998a*; *Burman and Ktistakis, 2010*; *Yang and Klionsky, 2010*; *Weidberg et al., 2011*; *Rubinsztein et al., 2012*).

A long-standing quest in the autophagy field has been to define the origin of the autophagosomal membrane. Recent data suggest a multi-membrane source model for autophagosome biogenesis. The endoplasmic reticulum (ER) supports PI3P-dependent formation of the omegasome, a cradle for phagophore generation and elongation (*Axe et al., 2008*; *Hayashi-Nishino et al., 2009*; *Yla-Anttila et al., 2009*). The outer membrane of the mitochondrion may also supply lipids for the phagophore and autophagosome (*Hailey et al., 2010*). Recently, a study by Hamasaki et al. (*Hamasaki et al., 2013*) indicates the ER–mitochondrial junction as being required for autophagosome biogenesis, possibly reconciling these two origins. In addition, clathrin-coated vesicles from the plasma membrane have been shown to promote phagophore expansion through the SNARE protein VAMP7 and its partner SNAREs (*Ravikumar et al., 2010*; *Moreau et al., 2011*). Moreover, ATG9-positive vesicles cycle between distinct cytoplasmic compartments to deliver membrane to a developing autophagosome or, in yeast, to phagophore assembly sites (PAS) (*Young et al., 2006*; *Sekito et al., 2009*; *Mari et al., 2010*; *Nair et al., 2011*; *Orsi et al., 2012*; *Yamamoto et al., 2012*). Autophagosomes may also acquire membrane from other sources including Golgi (*Geng et al., 2010*; *Ohashi and Munro, 2010*; *Yen et al., 2010*; *van der Vaart et al., 2010*), early endosomes (*Longatti et al., 2012*) and vesicles budding from the ER and Golgi (*Hamasaki et al., 2003*; *Zoppino et al., 2010*; *Guo et al., 2012*). Although tremendous progress has been made, a direct functional link between a membrane source and autophagosome biogenesis has not been established. Furthermore, the identity of the membrane determinant that responds to a stress signal to initiate autophagosome formation is unknown.

A variety of visual techniques have been developed to define the origin of the autophagosome membrane. Here, we developed a functional approach relying on the lipidation of LC3 to assay an early stage in autophagosome biogenesis. We establish a cell-free system that reflects many of the physiological and biochemical landmarks of early events in the autophagic pathway and define the ER–Golgi intermediate compartment (ERGIC), a membrane compartment between ER and Golgi for cargo sorting and recycling (*Appenzeller-Herzog and Hauri, 2006*), as a key membrane determinant for autophagosome biogenesis.

## Results

### Establishment of a regulated in vitro LC3 lipidation assay

A key step in autophagosome biogenesis is the generation of PE-lipidated LC3 by a ubiquitin-like conjugation system (*Ichimura et al., 2000*; *Kabeya et al., 2000*). The level of LC3 lipidation has long been a reliable measure of autophagy activity in vivo (*Klionsky et al., 2012*). In vitro LC3 lipidation has recently been reconstituted with synthetic liposomes, recombinant LC3 and other components including the ATG12–ATG5 conjugate, ATG7 and ATG3 (*Sou et al., 2006*; *Hanada et al., 2007*; *Oh-oka et al., 2008*; *Shao et al., 2007*; *Gao et al., 2010*). We sought to capture this modification in a more physiological context by relying on native membranes and cytosol to provide the core components of LC3 lipidation as well as any regulatory proteins that may be required for early autophagosome formation.

To establish such an assay, we mixed autophagosome precursor-deficient membranes with cytosol from normal and starved cells. Cells lacking ATG5 are deficient in starvation-induced autophagy and phago-phore formation (*Mizushima et al., 2001*). Hence they only contain unmodified LC3 (LC3-I) in both cytosol and membrane fractions (*Figure 1A*). Cytosols derived from WT cells (including WT MEF [mouse embryonic fibroblast], COS-7 [*Cercopithecus aethiops* fibroblast-like kidney cells], and HEK293T [human embryonic kidney 293T cells]) were highly enriched in LC3-I whereas the lipidated form of LC3 (LC3-II) sedimented with membranes (*Kabeya et al., 2000* and *Figure 1A*). We incubated membranes from *Atg5* KO MEFs with cytosol from WT MEFs in the presence of GTP and an ATP regeneration system (*Figure 1B*) and observed the formation of LC3-II in a time- (*Figure 1B*) and ATP-dependent manner (*Figure 1C*).

We compared the fractionation and biochemical properties of the in vitro-generated LC3-II to its in vivo counterpart. In a crude fractionation study, we found that the in vitro-generated LC3-II partitioned in the 16,000×*g* membrane fraction (*Figure 1—figure supplement 1A*). Moreover, the in vitro product resisted extraction with urea or $Na_2CO_3$ (*Figure 1—figure supplement 1B*) and was delipidated to LC3-I by ATG4B (*Figure 1D*), a cysteine protease that cleaves the C-terminal tail of LC3 and removes PE from LC3-II (*Tanida et al., 2004*). These properties are shared with LC3-II generated in vivo (*Kabeya et al., 2000*; *Tanida et al., 2004*).

Starvation-induced lipidation of LC3 requires the ATG12–ATG5 conjugate (*Mizushima et al., 2001*). To test the ATG5 dependence and starvation effect on in vitro LC3 lipidation, we incubated cytosols from either untreated or starved WT cells or *Atg5* KO MEFs with the corresponding membranes from *Atg5* KO MEFs (*Figure 2A*). LC3-II formation was stimulated about threefold in incubations containing cytosol from starved WT MEFs and membranes from starved *Atg5* KO MEFs, compared to incubations containing cytosol and membranes from non-starved MEFs (*Figure 2A*). Cytosol from *Atg5* KO MEFs did not generate LC3-II when combined with membranes from *Atg5* KO MEFs (*Figure 2A*). In addition, cytosols from COS-7 and HEK293T cells also reconstituted starvation-induced lipidation of LC3 (*Figure 2—figure supplement 1*). These data suggest that the cell-free LC3 lipidation is regulated by starvation-induced components in cells and is dependent on ATG5.

To test the physiological relevance of the cell-free reaction, we examined the effect of inhibitors of autophagy on the lipidation of LC3 in vitro. Starvation-induced autophagosome biogenesis requires the class III PI3K complex which contains ATG14, BECN1, VPS15, and the PI3K subunit VPS34 (*Burman and Ktistakis, 2010*; *Obara and Ohsumi, 2011*). Inhibition of the PI3K activity prevents autophagy. LC3 lipidation was inhibited in a dose-dependent manner by 3-methyladenine (3-MA) and wortmannin, two PI3K inhibitors of different potency but which act in the same concentration ranges to block autophagy in intact cells (*Figure 2B* and *Klionsky et al., 2012*).

In starved cells, downstream effector proteins recognize the PI3P generated by the autophagy-specific VPS34 PI3 kinase. The FYVE domain binds to PI3P (*Stenmark and Aasland, 1999*) and when expressed in excess blocks autophagy in the cell by sequestering PI3P (*Axe et al., 2008*). To study the

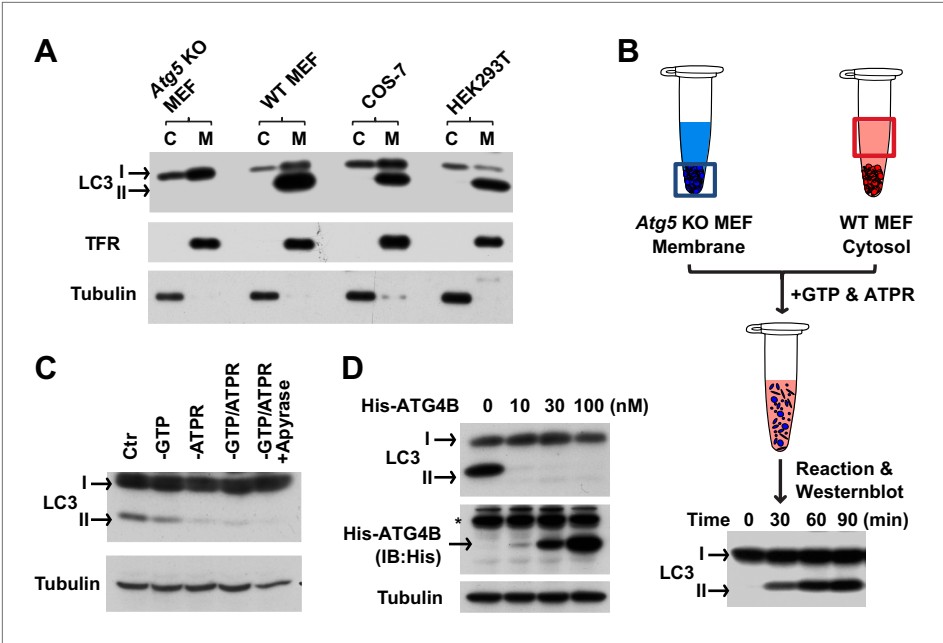

**Figure 1**. In vitro reconstitution of endogenous LC3 lipidation. (**A**) The distribution of LC3-I and LC3-II between the cytosol (C) and membrane (M) fractions from indicated cells. Cytosol and membranes from indicated cells were separated and evaluated by immunoblot (IB) with indicated antibodies. TFR, transferrin receptor (**B**) cell-free reconstitution of LC3 lipidation. Membranes from *Atg5* knockout (KO) MEFs were incubated with cytosol from wild type (WT) cells plus GTP and an ATP regeneration system (ATPR) for the indicated times. Then SDS-PAGE and immunoblot were performed to detect the generation of lipidated LC3 (LC3-II). (**C**) ATP dependence of in vitro LC3 lipidation. Reactions similar to (**B**) were performed in the absence or presence of indicated reagents followed by SDS-PAGE and immunoblot. (**D**) Delipidation of LC3 by ATG4B. A reaction similar to (**B**) was performed and the 16,000×*g* membranes were sedimented and solubilized with 1% TritonX-100. The indicated concentrations of ATG4B were incubated with the samples for 30 min followed by SDS-PAGE and immunoblot. Asterisk, non-specific band.

The following figure supplements are available for figure 1:

**Figure supplement 1**. Characterization of the in vitro-lipidated LC3.

role of PI3P in the in vitro reaction, we isolated a FYVE domain derived from FENS-1 (WDFY1), an endosomal protein (***Ridley et al., 2001***; ***Axe et al., 2008***), and included the peptide in a lipidation reaction mixture (***Figure 2—figure supplement 2A,B***; ***Figure 2C***). As reported in intact cells, the FYVE domain peptide inhibited LC3 lipidation in a dose-dependent manner whereas a cysteine to serine (C/S) mutation, which abolishes the ability of FYVE to bind PI3P (***Figure 2—figure supplement 2A,C*** and ***Axe et al., 2008***), had no effect on lipidation (***Figure 2C***).

One technical limitation is that the lipidation reaction relies on the conversion of endogenous LC3-I to LC3-II. In order to control the level of substrate, we isolated tagged recombinant LC3 expressed in *Escherichia coli*. LC3 is synthesized as a precursor that is processed by ATG4 cleavage to expose a glycine at position 120, the site of PE attachment (***Tanida et al., 2004***). Cell-free lipidation of recombinant T7-LC3 (aa1-120) required the glycine at position 120 and responded in a cytosol- and membrane concentration-dependent manner (***Figure 3—figure supplement 1A,B and C***). Lipidated T7-LC3 sedimented along with membranes at 16,000×*g*, a property shared with the endogenous LC3-II generated in vitro (***Figure 3—figure supplement 1D*** and ***Figure1—figure supplement 1A***).

We next evaluated the physiological requirements for lipidation using the T7-LC3 substrate. Cytosol and membranes isolated from starved cells stimulated T7-LC3 lipidation in vitro (***Figure 3A***), just as we observed with endogenous LC3 (***Figure 2A***). Likewise, PI3K inhibitors, 3-MA and wortmannin, and the PI3P blocking peptide, FYVE, blocked in vitro T7-LC3 lipidation in a dose-dependent manner (***Figure 3B,C***). Furthermore, cytosol deficient in ATG7, ATG3 or ATG5, key factors in the ubiquitin-like pathway for LC3 lipidation, failed to generate lipidated T7-LC3 in vitro (***Figure 3D,E and F***).

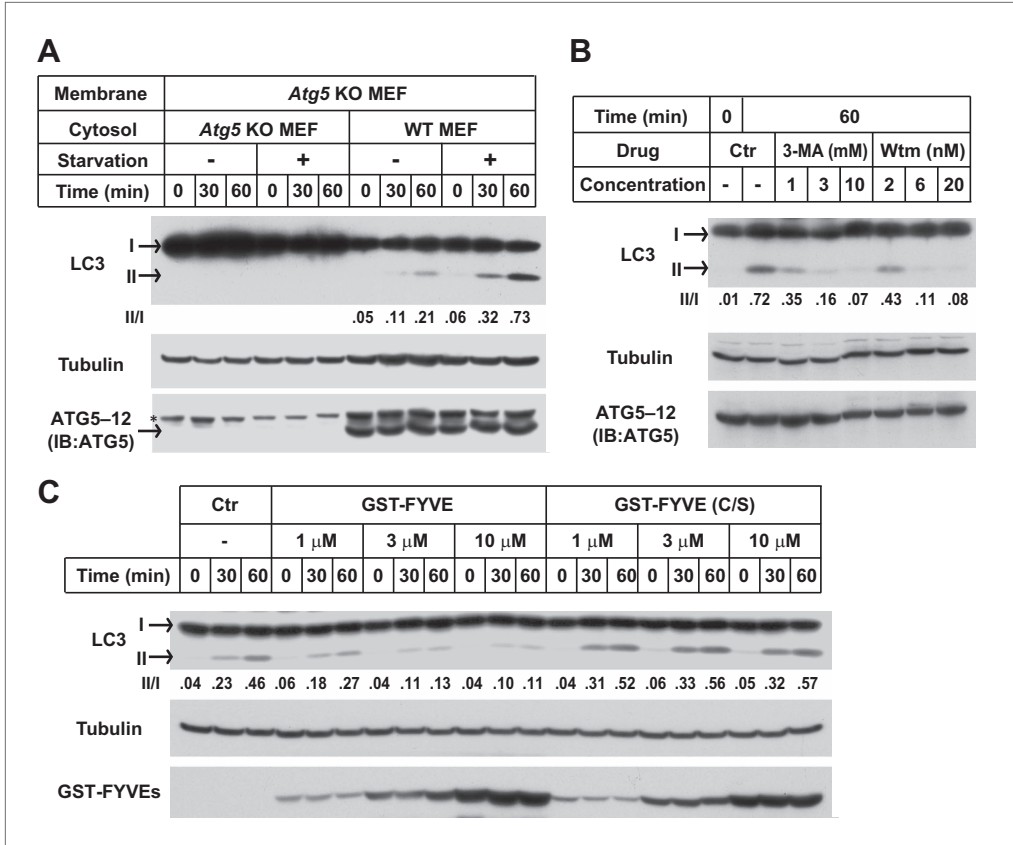

**Figure 2**. The in vitro lipidation of LC3 is regulated by ATG5, starvation and PI3K. (**A**) Starvation-promoted and ATG5-dependent lipidation of LC3. Indicated cells were either untreated or starved for 30 min. The in vitro lipidation reaction with the indicated combination of cytosols and membranes was performed. The formation of LC3-II was analyzed by SDS-PAGE and immunoblot. Asterisk, non-specific band (**B**) PI3K inhibitors 3-methyladenine (3-MA) and wortmannin (Wtm) inhibit LC3 lipidation. The in vitro lipidation reaction, with cytosol from starved WT MEFs and membrane from *Atg5* KO MEFs, was performed in the absence or presence of the indicated concentrations of 3-MA and wortmannin for 60 min. LC3 lipidation was analyzed by SDS-PAGE and immunblot. (**C**) PI3P dependence of in vitro LC3 lipidation. The in vitro lipidation reaction similar to (**B**) was performed in the absence or presence of increasing concentrations of GST-FYVE or FYVE (C/S) proteins for the indicated times. SDS-PAGE and immunoblot were performed to analyze the level of LC3-II. Quantification of lipidation activity was shown as the ratio of LC3-II to LC3-I (II/I).

The following figure supplements are available for figure 2:

**Figure supplement 1**. Starvation-promoted lipidation of LC3 by COS-7 or HEK293T cytosol.

**Figure supplement 2**. Purification and verification of GST-FYVEs.

Starvation induces autophagy through an MTORC1-ULKI protein kinase regulatory scheme (*Kim et al., 2011*). We found that cytosol from starved or untreated *Ulk1* KO MEFs reduced lipidation two to threefold relative to cytosol from WT cells (*Figure 3G*). Furthermore, T7-LC3 lipidation was stimulated two to threefold by two MTOR inhibitors, rapamycin (*Heitman et al., 1991*) and Torin 1 (*Liu et al., 2010*), known to induce autophagy (*Figure 3H,I*). Thus, for endogenous and recombinant LC3, the cell-free reaction reflects and responds to the major regulatory pathways of autophagy.

## Identification of the ERGIC as the membrane determinant that triggers in vitro LC3 lipidation

We employed the cell-free reaction as an assay to isolate the membrane responsible for LC3 lipidation. For this purpose, we devised a three-step membrane fractionation procedure and monitored enrichment of the lipidation activity with respect to a variety of membrane marker proteins and the lipid donor PE, in relation to a bulk membrane marker, phosphatidylcholine (PC) (*Figure 4*).

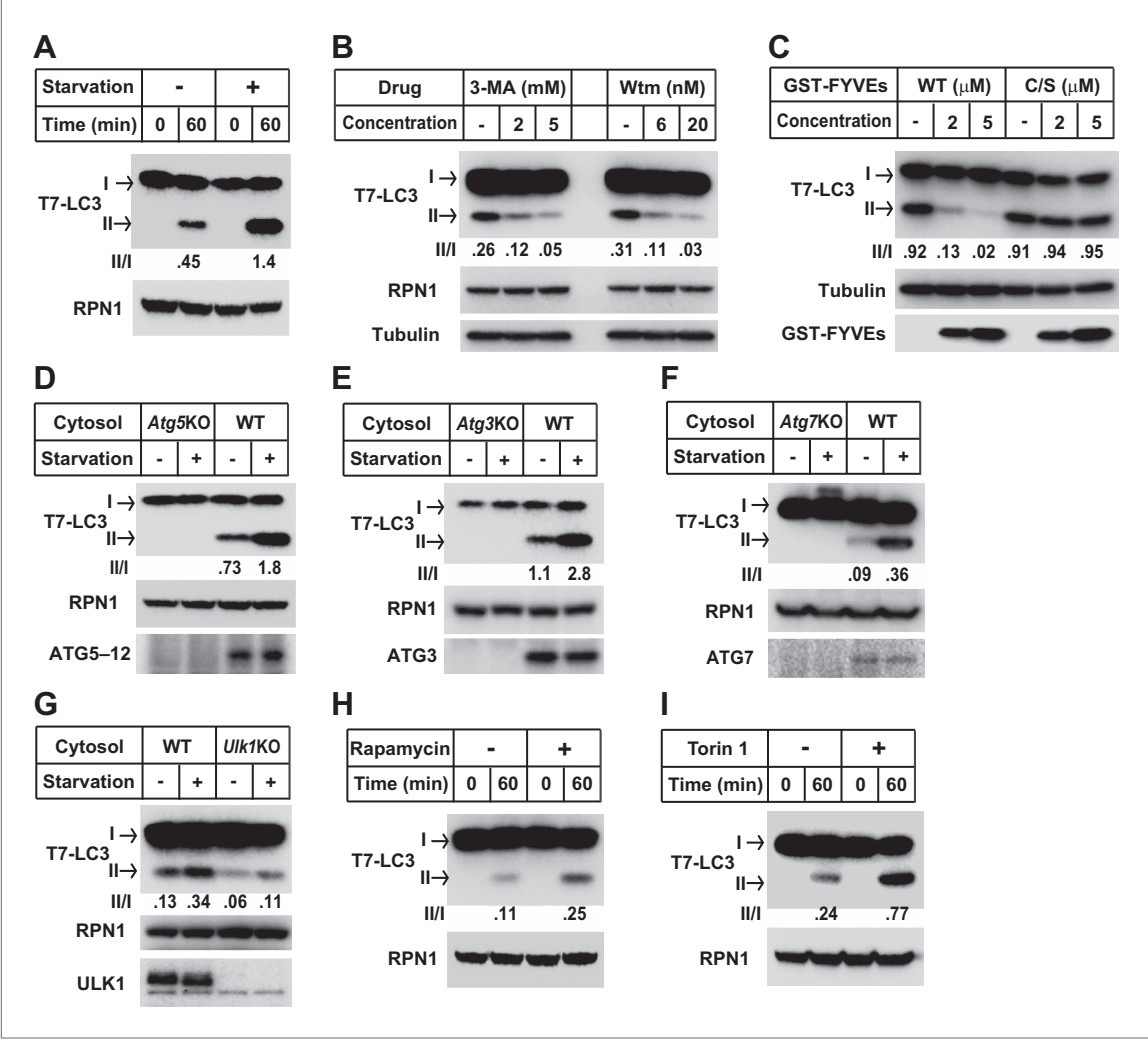

**Figure 3.** Recapitulation of the major regulatory pathways for autophagy by in vitro lipidation of T7-LC3. (**A**) Starvation-induced lipidation of T7-LC3. HEK293T and *Atg5* KO MEF cells were either untreated or starved for 90 min. The in vitro lipidation reaction was performed by incubating T7-LC3 with HEK293T cytosols and *Atg5* KO MEF membranes with indicated treatments for the indicated times followed by SDS-PAGE and immunoblot. (**B**) 3-methyladenine and wortmannin inhibit T7-LC3 lipidation. The in vitro lipidation reaction was performed by incubating T7-LC3 with cytosol from starved HEK293T and *Atg5* KO MEF membranes in the absence or presence of the indicated drugs for 60 min followed by SDS-PAGE and immunoblot. (**C**) PI3P dependence of in vitro T7-LC3 lipidation. In vitro lipidation reactions similar to (**B**) were performed in the absence or presence of the indicated concentrations of GST-FYVEs for 60 min followed by SDS-PAGE and immunoblot. (**D**) Dependence on ATG5 for T7-LC3 lipidation. The in vitro lipidation reaction was performed by incubating T7-LC3 with starved cytosols as indicated and *Atg5* KO MEF membranes for 60 min followed by SDS-PAGE and immunoblot to analyze LC3-II in the membrane fraction. (**E**) Dependence on ATG3 for T7-LC3 lipidation. A similar experiment was performed using cytosols from *Atg3* KO and WT MEFs, and membrane from Atg3 KO MEFs. (**F**) Dependence on ATG7 for T7-LC3 lipidation. A similar experiment was performed using cytosols from *Atg7* KO and WT MEFs, and membrane from *Atg7* KO MEFs. (**G**) Dependence on ULK1 for starvation-induced T7-LC3 lipidation. The in vitro lipidation reaction was performed by incubating T7-LC3 with untreated or starved cytosols as indicated and *Ulk1* KO MEF membranes for 60 min followed by SDS-PAGE and immunoblot of the membrane fraction. (**H**) Rapamycin-induced lipidation of T7-LC3. Cells were treated with 1 µM rapamycin or a control solution for 2 hr and cytosol was incubated with membranes as in (**A**). (**I**) Torin 1-induced lipidation of T7-LC3. Cells were treated with 200 nM Torin 1 or a control solution for 90 min and incubated with membranes as above. Quantification of lipidation activity was shown as the ratio of LC3-II to LC3-I (II/I).

The following figure supplements are available for figure 3:

**Figure supplement 1**. Purification of the T7-tagged LC3 and characterization of the lipidation.

In order to separate cellular membranes, we first performed differential centrifugation to obtain four membrane pellets containing different membrane markers (*Figures 4 and 5*). The PC level of each fraction was measured and normalized so that the lipidation activity for equal amounts of PC (specific activity) could

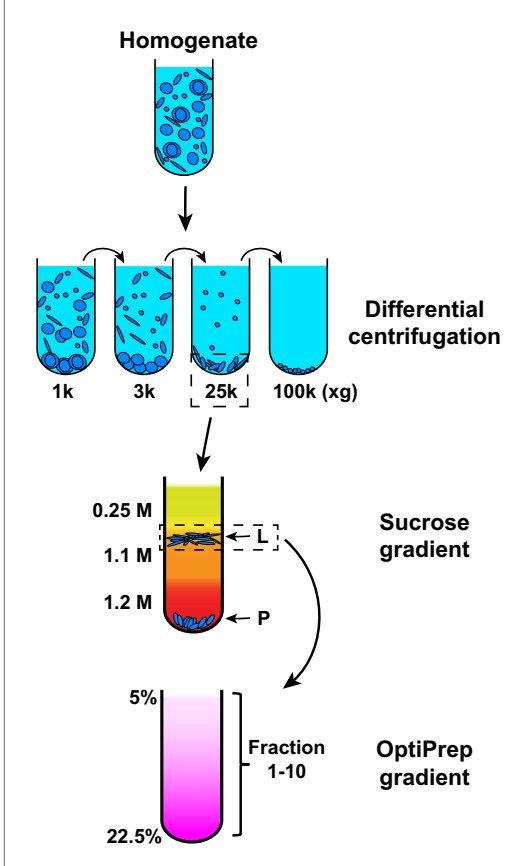

**Figure 4**. Membrane fractionation scheme. Briefly, *Atg5* KO MEFs were homogenized and the lysates were subjected to differential centrifugations with indicated g forces. The ability of each fraction to trigger T7-LC3 lipidation was examined. The 25,000×*g* (25k) pellet, which had the most activity, was selected and a sucrose gradient ultracentrifugation was performed to separate the 25k pellet to L (light) and P (pellet) fractions. The L fraction, which contained the majority of the activity to promote T7-LC3 lipidation, was further resolved on an OptiPrep gradient after which ten fractions from the top were collected and the lipidation activity was examined in each.

be determined (*Figure 5A,B and C*). The 25k and 100k fractions had significant lipidation activity (*Figure 5A,B*). The total activity contained in each fraction was calculated by multiplying the specific activity by the PC level (*Figure 5D*). The 25k pellet contained most (>70%) of the total activity, whereas the 100k pellet had little (<10%) total activity due to low levels of membrane. The 25k membrane was enriched in peroxisomes (PMP70/ABCD3), late endosomes (LAMP2) and cis-Golgi (GM130/GOLGA2). This fraction also contained membranes from the ERGIC (SEC22B and ERGIC53/LMAN1), plasma membrane/early endosomes (PM/Endo, TFR), ER (RPN1), ER exit sites (ERES, active sites on the ER that generate COPII-coated vesicles, SEC12), lysosomes (Cathepsin D), and ATG9 vesicles. Low levels of a mitochondrial marker (Prohibitin 1) and almost no trans-Golgi (TGN38/TGOLN2) or nuclear (Histone H4) compartments (*Figure 5A*) were detected.

To further fractionate the 25k membrane, we performed sucrose step gradient ultracentrifugation (*Figure 4*). This separated the 25k membrane to two distinct fractions, a light (L) fraction between the 1.1 M and 0.25 M layers of sucrose, and a pellet (P) fraction that sedimented to the bottom (*Figure 4*). The specific and total activity was determined as described above (*Figure 6*). Interestingly, the lipidation activity was almost exclusively retained in the L fraction, which was enriched in ERGIC, cis-Golgi, ATG9 vesicles and plasma membrane/early endosomes (*Figure 6A,B and D*). In contrast, the P fraction, which mainly consisted of ER, ERES, mitochondria, lysosomes and peroxisomes, induced very little T7-LC3 lipidation (*Figure 6A,B and D*).

To further refine the membrane source of T7-LC3 lipidation activity, we centrifuged the L fraction on an OptiPrep gradient and collected ten fractions from the top (*Figure 4*). The lipidation activity was distributed in fractions two through four which co-distributed with SEC22B and ERGIC53 (*Figure 7*), two ERGIC markers (*Zhang et al., 1999*; *Appenzeller-Herzog and Hauri, 2006*). Intriguingly, PE, the substrate for LC3 lipidation, was not enriched in the fractions that triggered T7-LC3 lipidation (*Figure 7B*). The high activity of these membrane fractions was not caused by selective enrichment of the autophagic factors directly contributing to LC3 lipidation, as all of these factors are enriched in the cytosol, or by influencing the formation of the ATG5–12–16 complex essential for LC3 lipidation (*Fujita et al., 2008*) compared with other membrane fractions (*Figure 7—figure supplement 1*). These data indicate that factors other than those directly involved in catalyzing LC3 lipidation contribute to the high lipidation activity of these membranes. We further found that the lipidation reaction triggered by the ERGIC-enriched membranes was enhanced by cytosol from starved cells and was inhibited by wortmannin and FYVE peptide (*Figure 7—figure supplement 2*). These data suggest that the lipidation activity of the isolated membranes is controlled by the pathway(s) that regulate autophagy in vivo.

The data from membrane fractionation indicate that an ERGIC-enriched membrane fraction induces LC3 lipidation. To determine if ERGIC is indeed the essential membrane for LC3 lipidation, we immunodepleted ERGIC from the L fraction with an anti-SEC22B antibody (*Figure 8A*). After

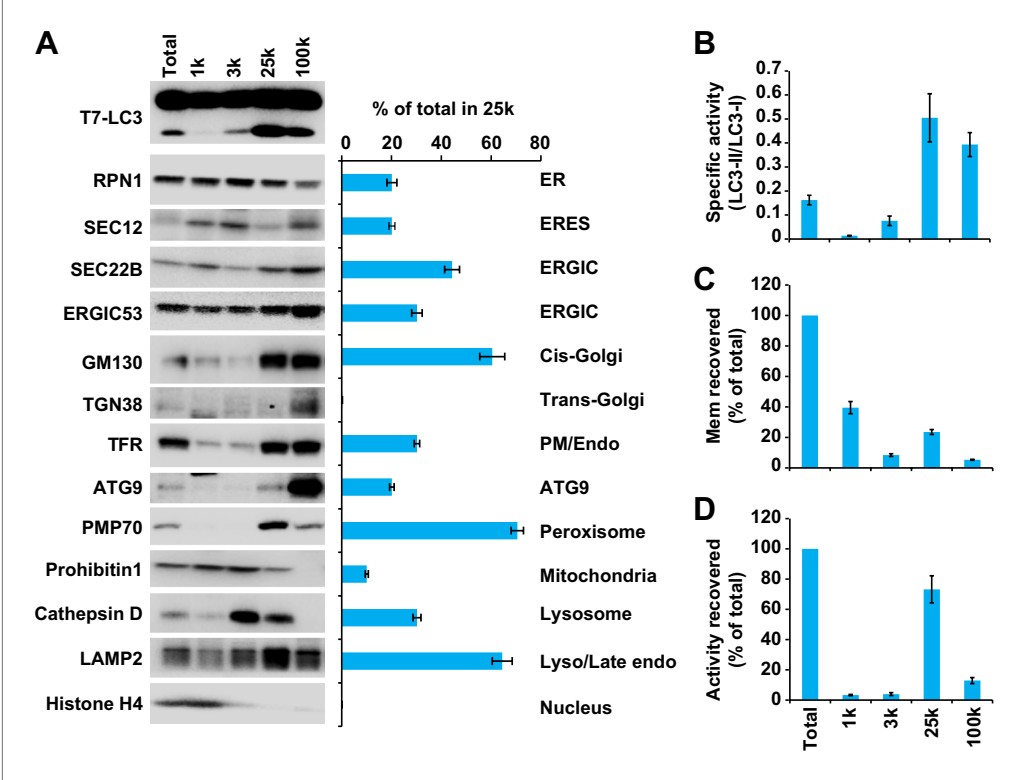

**Figure 5**. Separation of the total membrane by differential centrifugations. (**A–D**) A differential centrifugation experiment was performed as depicted in *Figure 4*. The total PC of each fraction was measured and presented as a percentage of the total membrane (**C**) and adjusted to a concentration of 0.6 mg/ml. The T7-LC3 lipidation activity of each fraction was tested and immunoblot was performed to examine the generation of lipidated T7-LC3 as well as the distribution of the indicated membrane markers (**A**). The level of each marker in the 25k pellet fraction was calculated as a percentage of the total membrane (**A**). The specific activity (the ability of each membrane fraction to induce LC3 lipidation with the equal amount of PC) of each membrane fraction to trigger T7-LC3 lipidation was measured as a ratio of lipidated to unlipidated T7-LC3s (**B**). The total activity recovered from each fraction was calculated by multiplying the specific activity by the corresponding PC level of each fraction and shown as a percentage of the total membrane (**D**). Error bars represent standard deviations of at least three experiments. RPN1, Ribophorin1; TFR, Transferrin receptor; Mem, membrane; Endo, endosome.

immunodepletion, both SEC22B and ERGIC53 membranes were reduced, whereas the plasma membrane/endosome membranes, indicated by transferrin receptor, were not affected. Significantly, the ability of the L fraction to induce T7-LC3 lipidation was reduced more than threefold (*Figure 8A*), suggesting that ERGIC contributes to the high activity of LC3 lipidation in the L fraction.

To test the direct role of ERGIC on LC3 lipidation, we immunoisolated SEC22B or KDEL receptor (KDELR, another ERGIC marker [*Capitani and Sallese, 2009*])-positive membranes (*Figure 8B,C*) for use in the in vitro lipidation assay. In both experiments, the immunoisolated membranes had substantially increased activity (about twofold or more than fivefold in the SEC22B or KDELR immunoisolated membranes, respectively) compared to the total input membrane (*Figure 8B,C*). In contrast, immunoisolation of lysosomes, plasma membrane/endosomes or ER did not enrich the lipidation activity compared to the total input membrane (*Figure 8D,E and F*). Therefore the data suggest that ERGIC is the most active membrane substrate for LC3 lipidation.

A recent report suggested that starvation induces the recruitment of ATG14 to a zone of adhesion between the ER and mitochondria, termed mitochondrial-associated endoplasmic reticulum membranes (MAM; *Hamasaki et al., 2013*). Hamasaki et al. suggested that this zone of adhesion might trigger the formation of the autophagosome (*Hamasaki et al., 2013*). Such a specialized patch of the ER could be an immediate precursor of the autophagosome. Markers of this adhesion were separated from the ERGIC fraction and the adhesion membrane isolated by the protocol of Wieckowski et al.

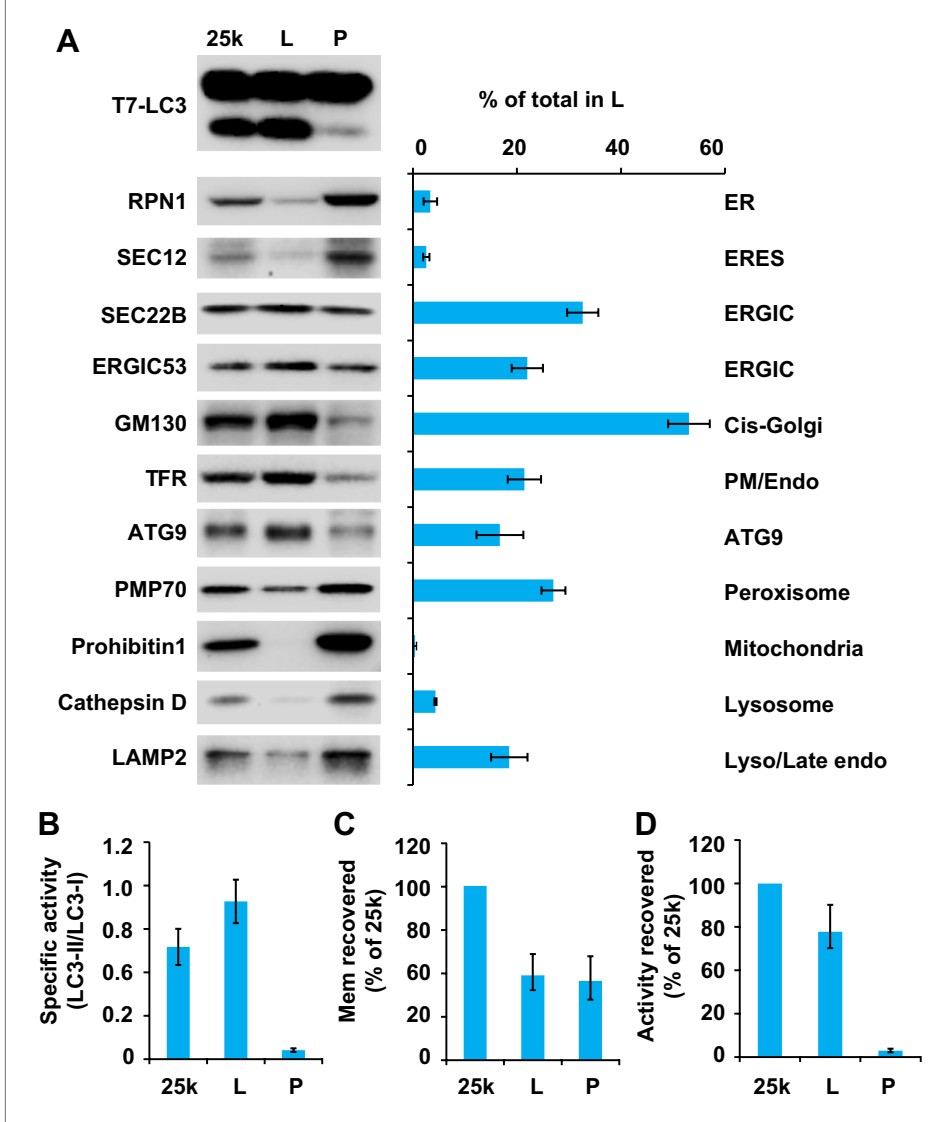

**Figure 6**. Separation of the 25k pellet fraction by sucrose gradient ultracentrifugation. (**A–D**) A sucrose step gradient ultracentrifugation to further separate the 25k pellet fraction was performed as depicted in **Figure 4**. The total PCs of each fraction were measured and presented as a percentage of the 25k pellet membrane (**C**) and adjusted to a concentration of 0.6 mg/ml. The T7-LC3 lipidation activities of the L and P fraction were tested and immunoblot was performed as in **Figure 5A**. The level of each marker in the L fraction was calculated as a percentage of the total membrane (**A**). The specific activity of each membrane fraction was measured as in **Figure 5B**. The total activity recovered from each fraction was calculated by multiplying the specific activity by the PC level of each fraction and shown as the percentage of 25k pellet membrane (**D**). Error bars represent standard deviations of at least three experiments.

(**Wieckowski et al., 2009**) contained little lipidation activity (**Figure 8—figure supplement 1**), suggesting this membrane alone may not be the direct template for LC3 lipidation. Consistent with Hamasaki et al., a fraction of ATG14 appeared on MAM in a starvation-stimulated manner (**Figure 8—figure supplement 1B**).

## ERGIC is a key membrane determinant for LC3 lipidation

To further test the importance of ERGIC in LC3 lipidation, we used inhibitors to deplete ERGIC in cultured cells. H89 is a protein kinase A (PKA) inhibitor that blocks COPII-coated vesicle assembly by preventing SAR1 binding to ER membrane when used at a high concentration (**Chijiwa et al., 1990**; **Aridor and Balch, 2000**). Brefeldin A (BFA) is a fungal metabolite that inhibits ARF1 activation

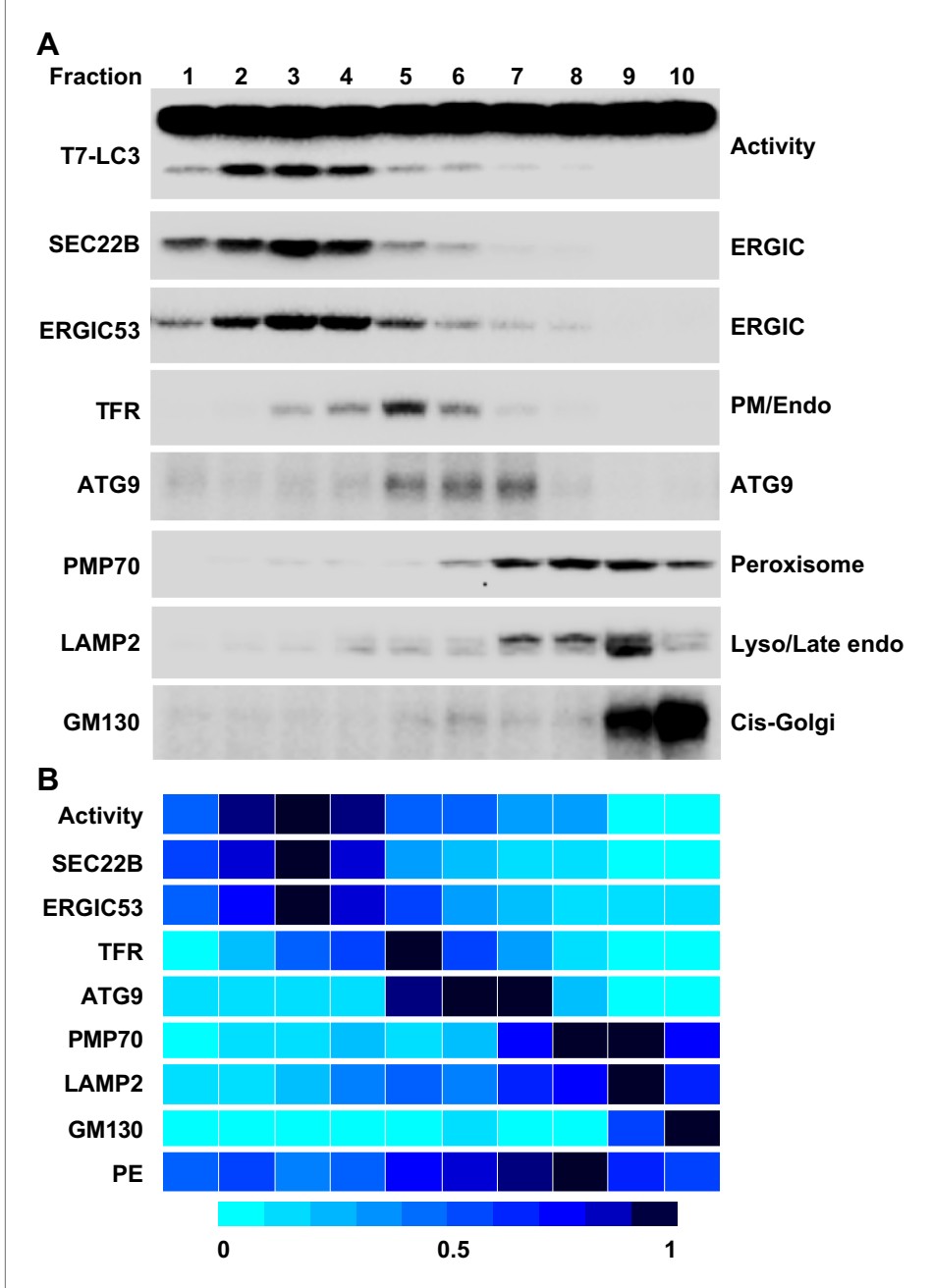

**Figure 7**. Separation of the L fraction by OptiPrep gradient ultracentrifugation. (**A–B**) An OptiPrep gradient ultracentrifugation was used to resolve membranes in the L fraction, as depicted in **Figure 4**. 10 fractions were collected. The total PCs of each fraction were measured and adjusted to a concentration of 0.6 mg/ml. The T7-LC3 lipidation activities of each fraction were tested and immunoblot was performed as in **Figure 5A**. The specific activity of each membrane fraction was measured similar to **Figure 5**. The PE level of each normalized fraction was determined. A heat map showing the relative levels of the specific activity, PE and each of the indicated markers was generated (**B**). In each group the fraction with the highest value was defined as 1.

The following figure supplements are available for figure 7:

**Figure supplement 1**. The ERGIC membrane promotes LC3 lipidation without altering cytosolic factors.

**Figure supplement 2**. The lipidation activity of the ERGIC-enriched fractions are regulated by starvation and PI3K.

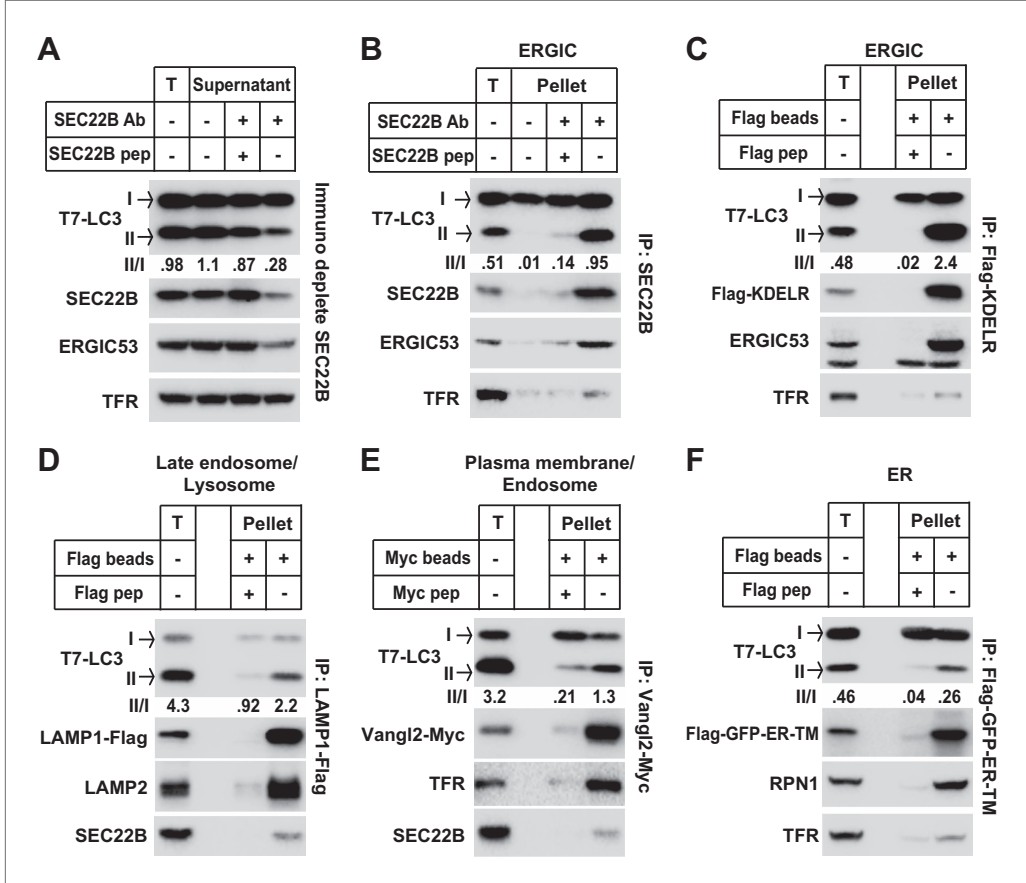

**Figure 8**. ERGIC directly triggers in vitro LC3 lipidation. (**A**) Immunodepletion of ERGIC membrane from L fraction reduces in vitro lipidation activity. The L fraction was prepared as shown in ***Figures 4 and 5***. An immunodepletion experiment with indicated combinations of anti-SEC22B antibody (Ab) and blocking peptide (pep) was performed. The membranes from the flow-through were collected and the in vitro lipidation reaction was performed. Equal amounts of membrane from each group were used for the lipidation reaction. T, total membrane from the L fraction. (**B**) Enrichment of lipidation activity on the SEC22B-positive membranes. Immunoisolation of SEC22B positive membranes from the L fraction of *Atg5* KO MEFs was performed and the in vitro lipidation reaction was conducted with membranes bound to the beads as well as the total membrane (T) from the L fraction. The total membrane used was adjusted to the same amount of the membranes (based on PC content) specifically bound to the beads in the reaction. (**C**) Enrichment of lipidation activity on KDEL Receptor (KDELR)-positive membranes. *Atg5* KO MEFs were transfected with a plasmid encoding the Flag-tagged KDELR protein. 48 hr after transfection, KDELR-positive membranes were immunoisolated with anti-Flag agarose and assayed for lipidation activity as in (**B**). (**D–F**) Lipidation activity was not enriched in late endosome/lysosome, plasma membrane/endosome or ER membranes. *Atg5* KO MEFs were transfected with plasmids encoding LAMP1-Flag (**D**), Vangl2-Myc (**E**) or Flag-GFP-ER-TM (**F**). 48 hr after transfection, the 25k pellet fractions (for LAMP1-Flag and Flag-GFP-ER-TM) or the L fraction were collected and immunoisolations were performed with anti-Flag agarose or anti-Myc agarose as described above and assayed for lipidation activity. Quantification of lipidation activity was shown as the ratio of LC3-II to LC3-I (II/I).

The following figure supplements are available for figure 8:

**Figure supplement 1**. Mitochondrial-associated endoplasmic reticulum membranes (MAM) are not active to trigger in vitro LC3 lipidation.

---

(***Peyroche et al., 1999***). Treatment of cells with a high concentration of H89 (100 µM) led to the dispersal of ERGIC53 whereas GM130, a cis-Golgi marker, remained in the perinuclear region, suggesting that the ERGIC is disrupted but not the cis-Golgi (***Figure 9—figure supplement 1***). A low concentration of H89 (10 µM), which is enough to inhibit PKA, did not affect the localization of either ERGIC53 or GM130 (***Figure 9—figure supplement 1***). BFA treatment collapsed the Golgi into puncta colocalizing

with ERGIC53 (*Figure 9—figure supplement 1*). Treatment of cells with 100 µM H89 after BFA treatment dispersed both ERGIC53 and GM130 (*Figure 9—figure supplement 1*).

To assess the membrane fractions for retention of lipidation activity, we treated cells with the indicated drugs and the total membrane from each sample was collected and incubated with cytosol from starved cells (*Figure 9*). Membrane from cells treated with a high but not a low concentration of H89 (100 µM vs 10 µM) lost the ability to activate LC3 lipidation (*Figure 9A*). Membranes from BFA-treated cells did not show diminished lipidation activity, nor did BFA mitigate or enhance the effect of H89 treatment on lipidation activity (*Figure 9A*). Clofibrate is a peroxisome-proliferator activated receptor (PPAR) agonist that inhibits ER-to-Golgi transport and promotes retrograde transport of Golgi vesicles back to the ER through an unknown mechanism independent of PPAR activation (*de Figueiredo and Brown, 1999*). Clofibrate treatment led to the dispersal of ERGIC53 and GM130 (*Figure 9—figure supplement 1*). Like H89, membranes from clofribate-treated cells failed to promote LC3 lipidation (*Figure 9B*). As controls, kbNB142-70, a PKD inhibitor (*Bravo-Altamirano et al., 2011*), had no affect and Pitstop 2, a clathrin inhibitor (*von Kleist et al., 2011*), only moderately decreased in vitro LC3 lipidation (*Figure 9B*).

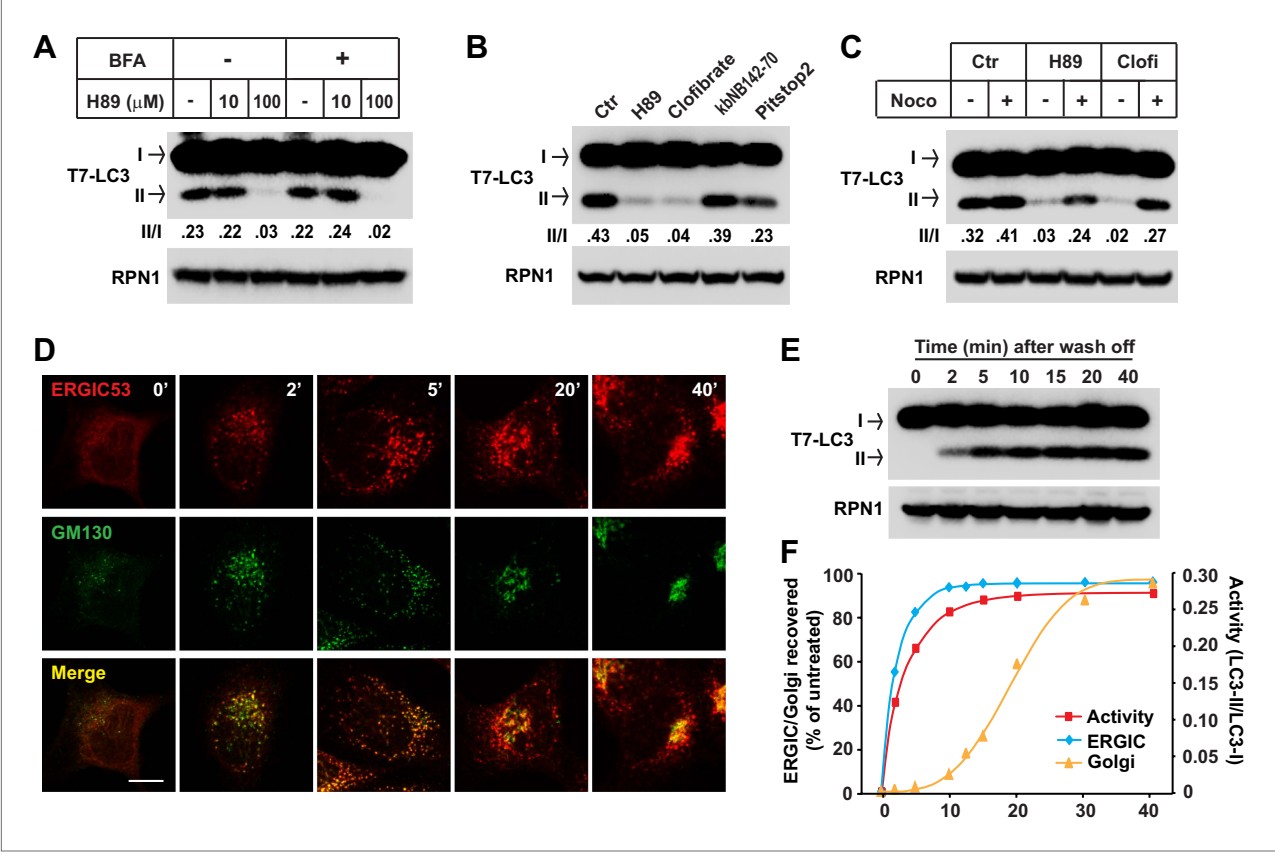

**Figure 9**. ERGIC is required for in vitro LC3 lipidation. (**A** and **B**) In vivo depletion of ERGIC abolishes the in vitro lipidation of LC3. *Atg5* KO MEFs were treated without or with 10 µg/ml Brefeldin A (BFA) for 30 min and then incubated with the indicated concentrations of H89 for 10 min (**A**). Alternatively, cells were directly treated with the indicated drugs: 100 µM H89, 500 µM clofibrate, 50 µM kbNB142-70 and 50 µM Pitstop2 (**B**). Total membranes from the cells were collected, the lipidation reaction with T7-LC3 was performed and the products evaluated by SDS-PAGE and immunoblot. Ctr, control (**C**) Blocking ERGIC disruption preserved the in vitro lipidation of LC3. *Atg5* KO MEFs were treated with control, H89 or clofibrate (Clofi) in the absence or presence of nocodazole (Noco). Lipidation reactions with the total membranes from the treated cells were performed. (**D–F**) The in vitro lipidation of LC3 recovers with restoration of ERGIC. *Atg5* KO MEF cells were treated with BFA followed by 100 µM H89. Cells were then washed with fresh medium to remove the drugs and, at indicated intervals, samples were collected for immunofluorescence (**D**) or total membrane collection for the in vitro lipidation reaction (**E**). Quantification of the recovery of lipidation activity, ERGIC and Golgi are shown in (**F**). Bar, 10 µm. Quantification of lipidation activity was shown as the ratio of LC3-II to LC3-I (II/I).

The following figure supplements are available for figure 9:

**Figure supplement 1**. Immunofluorescence showing the effect of indicated drugs on ERGIC and Golgi.

Retrograde transport to the ER requires microtubules. Disruption of microtubules by nocodazole inhibited the retrograde transport of ERGIC53 towards ER induced by H89 or clofibrate, preserving its punctate localization (*Figure 9—figure supplement 1*). Correspondingly, the inhibitory effects of H89 and clofibrate were reversed in respect to the cell-free lipidation activity (*Figure 9C*). To examine the reversibility of these effects, we treated cells with BFA and H89 for 20 min and then washed the cells into fresh medium for periods up to 40 min (*Puri and Linstedt, 2003*). ERGIC53 puncta began to reappear within 2 min and the normal localization was fully restored by 20 min. Cis-Golgi regeneration, indicated by the perinuclear accumulation of GM130, was slower and was completed by about 30 min (*Figure 9D,F*). Aliquots examined in the cell-free LC3 lipidation assay displayed a rapid return of activity correlating with the kinetics of ERGIC recovery (*Figure 9E,F*). These results support our membrane fractionation results concerning the role of the ERGIC in lipidation of LC3.

## ERGIC is required for autophagosome biogenesis

To test the role of ERGIC in autophagosome formation, we starved cells in the presence or absence of ERGIC-depleting drugs H89 and clofibrate (*Figure 10A*). We then monitored autophagosome biogenesis by immunofluorescence analysis of LC3 puncta formation (*Klionsky et al., 2012*). Under normal conditions, LC3 is dispersed in the cell, indicating a low level of autophagy (*Figure 10A*). Starvation-induced LC3 puncta formation was abolished by H89 (100 µM) and clofibrate (*Figure 10A,B*). Depressed formation of LC3 puncta was not due to enhanced autophagosome turnover because chloroquine, which blocks a late step in autophagy (*Klionsky et al., 2012*), did not mitigate the effect of H89 (100 µM) or clofibrate (*Figure 10A,B*). Puncta formation of another phagophore marker, ATG16 (*Fujita et al., 2008*), was also blocked by ERGIC depletion (*Figure 10—figure supplement 1*).

In addition to inhibiting vesicle traffic from the ER, H89 is a potent inhibitor of PKA. In order to determine whether the observed decrease in the number of LC3 puncta could be due to PKA inhibition and not loss of ERGIC, we tested a low concentration of H89 that inhibits PKA but has no effect on the ERGIC (*Figure 9—figure supplement 1*). In contrast to the effect of a high concentration of H89 (100 µM), cells treated with the low concentration (10 µM) showed a moderate increase in LC3 puncta with or without chloroquine (*Figure 10A,B*). A recent study also reported that PKA inhibition promotes autophagy (*Cherra et al., 2010*), thus PKA inhibition appears not to be the basis of the effect of H89 on autophagy.

As an additional test of the role of ERGIC in autophagosome formation, we introduced mutant forms of the SAR1A GTPase to inhibit the generation of COPII vesicles. A GTP-bound mutant of SAR1A (H79G) locks COPII membrane cargos on the ERES and a GDP-bound mutant, SAR1A T34N, completely blocks COPII coat formation (*Ward et al., 2001*). Overexpression of either SAR1A H79G or T34N led to dispersed ERGIC53 localization (*Figure 10—figure supplement 2*) and inhibited starvation-induced LC3 puncta formation in both control and chloroquine-treated cells (*Figure 10C,D*). We conclude that the ERGIC is a precursor of or contributes to the formation of the autophagosome.

## ERGIC recruits ATG14 and DFCP1, two early markers of autophagosome formation

ATG14 is the key mediator bridging upstream cytosolic signals and the autophagic membrane reorganization response. Upon starvation, ATG14 is recruited to a membrane along with the rest of the class III PI3K complex to generate PI3P (*Matsunaga et al., 2009*; *Sun et al., 2008*; *Zhong et al., 2009*; *Matsunaga et al., 2010*). These events can be visualized by the localization of ATG14 and DFCP1 to puncta in starved cells (*Axe et al., 2008*; *Matsunaga et al., 2010*). To test the role of ERGIC in this pathway, we treated starved cells with H89. As shown in *Figure 11*, 100 µM H89 but not 10 µM H89 prevented the formation of both ATG14 and DFCP1 puncta (*Figure 11A,B*). Similar inhibition was also observed in starved cells expressing the two SAR1A mutants, H79G and T34N (*Figure 11C,D*).

We developed a cell-free approach to measure the recruitment of ATG14 and DFCP1 to ERGIC membranes. *Atg5* KO MEFs were treated with or without H89 (100 µM). Cells were lysed and membranes from a post-nuclear supernatant fraction were mixed with cytosol from starved HEK293 cells transfected with tagged forms of ATG14 and DFCP1. The mixes were incubated in the presence of ATP and a regeneration system. Membranes were resolved on an OptiPrep buoyant density step gradient. We observed recruitment of tagged ATG14 and DFCP1 to a buoyant membrane fraction from untreated but much less from H89-treated cells (*Figure 12*). Recruitment was dependent on membranes, and stimulated by starvation (*Figure 12—figure supplement 1*). In starved cells, ATG14 acts upstream of PI3K activity whereas DFCP1 puncta formation requires PI3P generation

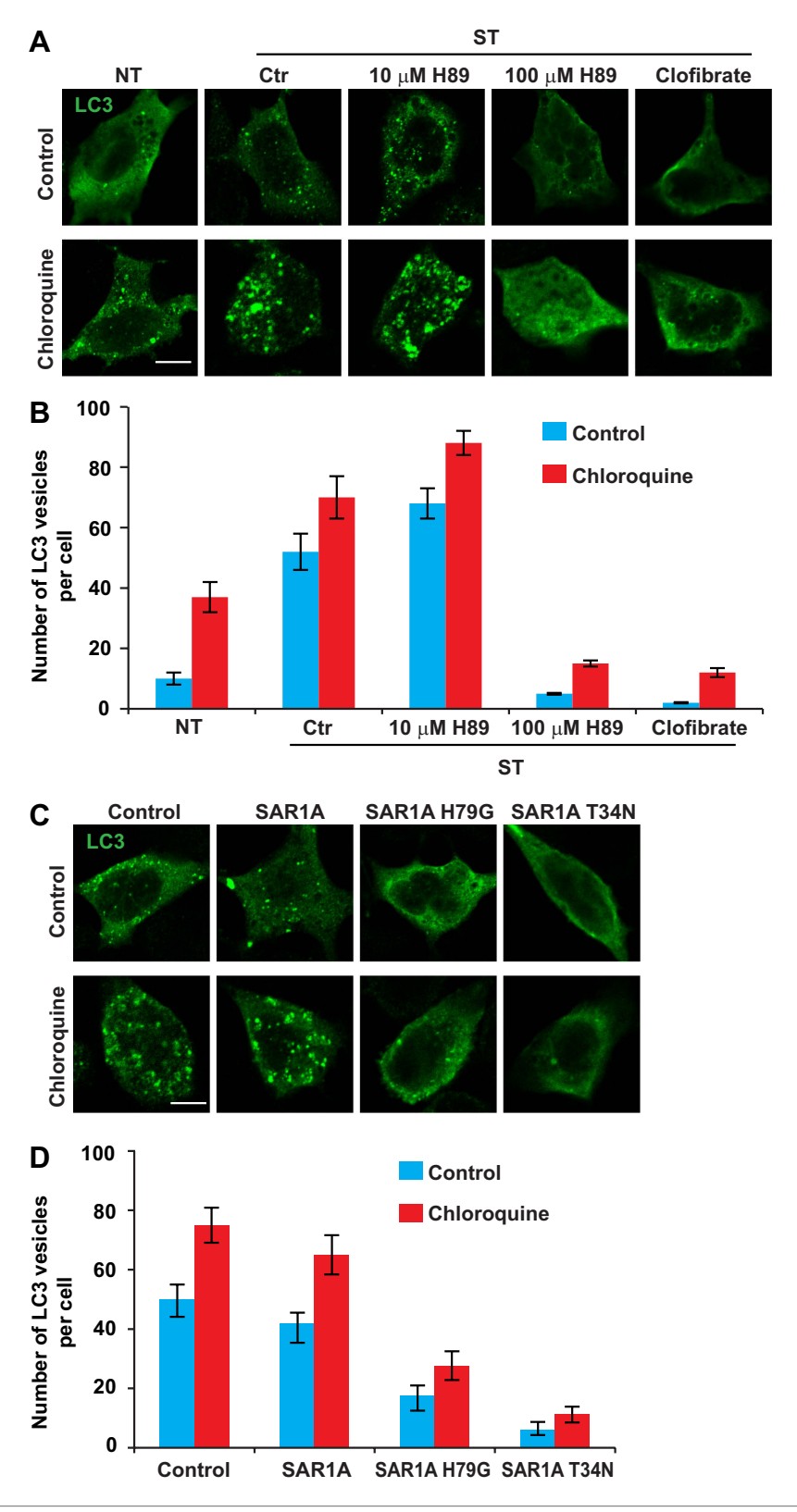

**Figure 10**. ERGIC is required for starvation-induced LC3 puncta formation. (**A**) Drugs that disrupt ERGIC inhibit LC3 puncta formation. MEFs were transfected with plasmids encoding Myc-LC3. After transfection (24 hr), the cells were either non-starved (NT) or starved (ST) in the absence or presence of the indicated drugs followed by
*Figure 10. Continued on next page*

*Figure 10. Continued*

immunofluorescence using anti-Myc antibody. Bar, 10 µm. (**B**) Quantification of the cells shown in (**A**). Error bars represent standard deviations of three experiments. (**C**) Genetically disrupting ERGIC inhibits LC3 puncta formation. MEF cells were co-transfected with plasmids encoding Myc-LC3 and the indicated SAR1A variants. After transfection (24 hr), the cells were starved in the absence or presence of chloroquine followed by immunofluorescence using anti-Myc antibody. Bar, 10 µm. (**D**) Quantification of the cells shown in (**C**). Error bars represent standard deviations of three experiments.

The following figure supplements are available for figure 10:

**Figure supplement 1**. Drugs that disrupt ERGIC inhibit starvation-induced ATG16 puncta formation.

**Figure supplement 2**. Effects of SAR1A variants on ERGIC.

(*Itakura and Mizushima, 2010*). Correspondingly, membranes from cells treated with PI3K inhibitors recruited ATG14 but not DFCP1 (*Figure 12—figure supplement 1B and C*). In addition to the membrane recruitment result, immunofluorescence studies showed colocalization of ATG14 and DFCP1 with ERGIC53 after starvation (*Figure 12—figure supplement 2*). We conclude that the ERGIC membrane is an early site for the assembly of proteins responsible for the formation of the phagophore membrane.

## Discussion

In this study, we have identified the ERGIC as a key membrane determinant in the biogenesis of autophagosomes. We first developed a cell-free assay based on in vitro LC3 lipidation to measure autophagosome biogenesis (*Figures 1–3*). By combining this assay with membrane fractionation, we identified an ERGIC-enriched fraction as the most active membrane to trigger LC3 lipidation (*Figures 4–7*). Next we used organelle immuno-/inhibitor depletion and immunoisolation to demonstrate that the ERGIC is necessary and sufficient to support LC3 lipidation (*Figures 8 and 9*). Finally, we provided evidence that the ERGIC membrane acts by recruiting ATG14 to initiate PI3K activity, an early step essential for autophagosome biogenesis (*Figures 10–12*).

Numerous morphological studies have indicated several possible sources of the autophagosomal membrane (*Burman and Ktistakis, 2010*; *Mari et al., 2011*; *Weidberg et al., 2011*; *Rubinsztein et al., 2012*). Indeed, it is improbable that one organelle contributes all the membrane constituents that become part of a mature autophagosome. Nonetheless, it seems likely that one membrane responds to the signal that triggers autophagosome formation and the identity of that membrane has remained elusive. Our isolation of the ERGIC as the locus of LC3 lipidation is in line with morphological studies that describe an omegasome structure projecting directly from the ER membrane (*Axe et al., 2008*; *Hayashi-Nishino et al., 2009*). However, our results show clearly that the bulk ER membrane is not the site of lipidation, thus if the omegasome arises from the ER, it must become modified in some way to be active for LC3 lipidation.

Starvation induces the membrane localization of soluble oligomeric proteins including ATG14 and the PI3K complex, followed by the recruitment of DFCP1 to generate the omegasome (*Axe et al., 2008*; *Matsunaga et al., 2009*; *Sun et al., 2008*; *Zhong et al., 2009*; *Matsunaga et al., 2010*). This process occurs upstream of phagophore generation (*Itakura and Mizushima, 2010*). Our data show that in starved cells and in isolated membranes, the presence of ERGIC is required for the efficient membrane recruitment of ATG14 and DFCP1 (*Figures 11 and 12*). Thus the ERGIC may play a role in an early stage of phagophore formation by providing a platform to recruit the class III PI3K complex and provide precursor membranes for phagophore initiation, which may be further expanded in a special subdomain of ER.

How and why the ERGIC is used to trigger phagophore formation remains unclear. Perhaps the tubular and curved structure of the ERGIC (*Appenzeller-Herzog and Hauri, 2006*) in mammalian cells favors recruitment of the ATG14 complex and subsequently of other components. ATG14 has been reported to sense membrane curvature via an amphipathic alpha helix located in a C-terminal 'BATS' domain (*Fan et al., 2011*). In yeast, it has been shown that highly curved membranes positive for ATG9 are delivered to the PAS (*Mari et al., 2010*; *Nair et al., 2011*; *Yamamoto et al., 2012*). Subsequently, the curvature sensing protein Atg1 recruits Atg13 and the Atg17–31–29 protein complex to initiate the formation of the phagophore (*Ragusa et al., 2012*). The suggested requirement for a tubular

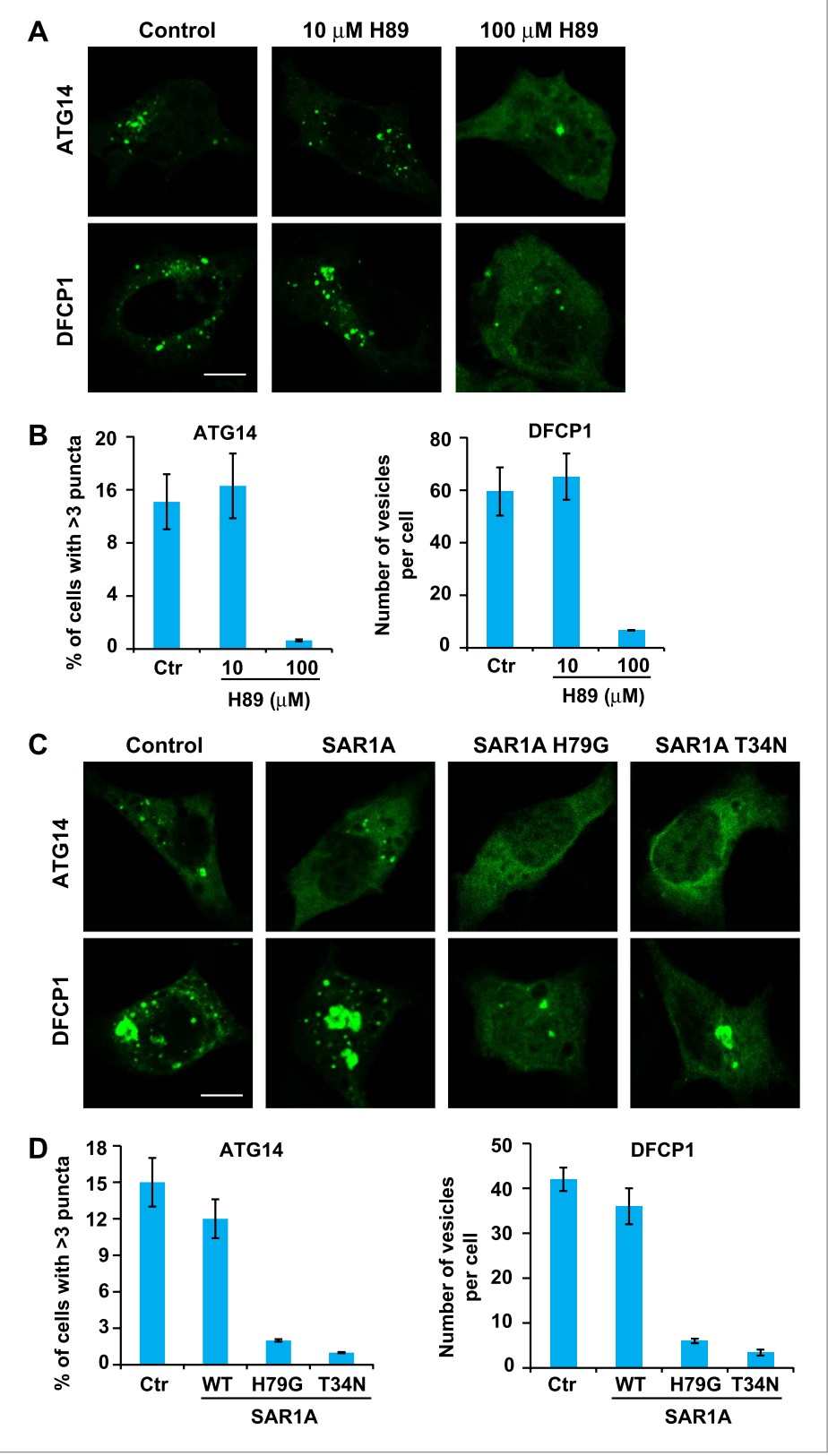

**Figure 11**. ERGIC is required for the starvation-induced localization of ATG14 and DFCP1 to puncta. (**A**) H89 inhibits ATG14 and DFCP1 puncta formation. MEF cells were transfected with plasmids encoding EGFP-tagged ATG14 or DFCP1. After transfection (24 hr), cells were starved in the absence or presence of the indicated concentrations of
*Figure 11. Continued on next page*

*Figure 11. Continued*

H89 followed by fixation and direct visualization of the EGFP signal. Bar, 10 µm. (**B**) Quantification of the cells shown in (**A**). Error bars represent standard deviations of three experiments. (**C**) Expression of SAR1A mutants inhibits the formation of puncta that contain ATG14 and DFCP1. MEF cells were co-transfected with plasmids encoding EGFP-tagged ATG14 or DFCP1 and indicated SAR1A-DsRed variants. After transfection (24 hr), cells were starved followed by fixation and direct visualization of EGFP signal. Bar, 10 µm. (**D**) Quantification of the cells shown in (**C**). Error bars represent standard deviations of three experiments.

membrane together with the possible existence of an integral membrane protein(s) that triggers ATG14 recruitment are now open for biochemical analysis.

The cell-free LC3 lipidation reaction responds to a starvation signal, likely originating in the cytosolic fraction. Fractionation of the cytosol should reveal the full range of biochemical requirements including regulatory components induced by starvation as well as the core proteins essential for LC3 lipidation. Furthermore, this approach could be exploited to evaluate the maturation of the phagophore through subsequent stages of morphological transformation including envelope closure and fusion with the lysosome.

## Materials and methods

### Materials, antibodies, and plasmids

We obtained horseradish peroxidase-conjugated goat anti-mouse or anti-rabbit IgG from Jackson ImmunoResearch Laboratories (West Grove, PA); fluorescent secondary antibodies and Earle's Balanced Salt Solution (EBSS) from Invitrogen (Grand Island, NY); PIP Strips from Echelon (Salt Lake City, UT); 3-methyladenine (3-MA), wortmannin, rapamycin, H89 and clofibrate from Sigma (St. Louis, MO); ATG4B from Boston Biochem (Cambridge, MA); SEC22B antibody blocking peptides (sequence CG+HSEFDEQHGKKVPTVSRPYSFIEFDT) from David King (University of California, Berkeley); Torin 1 and kbNB142-70 from Tocris (Minneapolis, MN); Pitstop 2 from Abcam (Cambridge, MA); reagents for PE measurement as described by Hokazono et al. (*Hokazono et al., 2011*) and other reagents from previously described sources (*Ge et al., 2008, 2011*). Amine oxidase was kindly provided by Eisaku Hokazono (Kyushu University, Higashi-ku, Japan).

Mouse anti-GM130, transferrin receptor, PMP70 and rabbit anti-Prohibitin-1, SEC22B and Ribophorin 1 antibodies were described before (*Ge et al., 2008*; *Schindler and Schekman, 2009*; *Ge et al., 2011*). We purchased mouse anti-Flag, rabbit anti-ERGIC53, anti-LC3, anti-LAMP2, anti-ULK1, anti-ATG14 and anti-BECN1 antibodies from Sigma (St. Louis, MO); mouse anti-T7 antibody from EMD (Billerica, MA); hamster anti-ATG9, mouse anti-tubulin, rabbit anti-FACL4 and HRP-labeled anti-GST antibodies from Abcam (Cambridge, MA); rabbit anti-Cathepsin D from Epitomics (Burlingame, CA); rabbit anti-TGN38

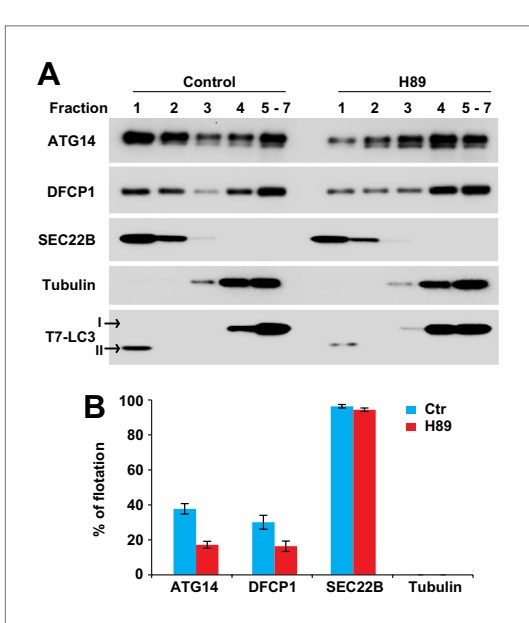

**Figure 12**. ERGIC is required for membrane recruitment of ATG14 and DFCP1. (**A**) Disruption of ERGIC inhibits membrane recruitment of ATG14 and DFCP1. *Atg5* KO MEFs were either untreated or treated with H89. Membranes were collected and incubated with cytosol of HEK293T cells expressing ATG14-HA and EGFP-DFCP1. A buoyant density flotation assay was performed followed by immunoblot. (**B**) Quantification of the floated markers shown in (**A**). The quantification of samples from the buoyant density gradient was calculated as the ratio of chemiluminescence in the first two fractions to the sum of all fractions. Error bars represent standard deviations of three experiments.

The following figure supplements are available for figure 12:

**Figure supplement 1**. Establishment of the in vitro membrane recruitment assay.

**Figure supplement 2**. Atg14L and DFCP1 puncta colocalize with ERGIC.

from Novus Biologicals (Littleton, CO); mouse anti-ATG7, anti-ATG3, anti-ATG5 and anti-ATG16 from MBL (Woburn, MA); goat anti-SEC12 antibody from R&D Systems (Minneapolis, MN); mouse anti-Myc antibodies from Cell Signaling (Boston, MA); mouse anti-GFP antibody from Santa Cruz (Dallas, Texas); rabbit anti-Histone H4 antibody from the Robert Tjian lab (University of California, Berkeley).

The plasmids encoding ATG14-EGFP, ATG14-HA, Myc-LC3 and ATG16-Myc were kindly provided by Qing Zhong lab (University of California, Berkeley). The EGFP-DFCP1 plasmid was kindly provided by Nicholas Ktistakis lab (Babraham Institute, UK). And the LAMP1-RFP-Flag plasmid was from Addgene (provided by the Sabatini lab, Whitehead Institute). The GST-FYVE, GST-FYVE(C/S) and T7-LC3 plasmids were constructed by subcloning the indicated inserts from the GFP-TM-FYVE, GFP-TM-FYVE(C/S) constructs (Nicholas Ktistakis, Babraham Institute, UK) and Myc-LC3 plasmids into pGEX4T1 and pET28a vectors. The encoded proteins were expressed in *E. coli* BL21 and affinity purified with glutathione or Ni Sepharose (GE Healthcare Life Sciences, Piscataway, NJ). The Flag-GFP-ER-TM plasmid was generated by PCR insertion of a Flag tag into the GFP-ER-TM plasmid (Nicholas Ktistakis, Babraham Institute, UK). The human Vangl2-myc plasmid was described before (*Guo et al., 2013*). Inserts from the pGEX-Sar1As (*Kim et al., 2005*) were subcloned into the DsRed-Monomer-N1 vector to generate DsRed-tagged SAR1A plasmids.

## Cytosol preparation

The cells were cultured to confluence and either untreated or starved in EBSS (for lipidation using endogenous LC3, MEF cells were starved for 30 min while HEK293T and COS-7 cells for 1 hr; for lipidation using T7-LC3, HEK293T cells were starved for 1.5 hr). Then the cells were harvested by scraping and centrifuging at $600 \times g$ for 5 min, washed with PBS followed by another $600 \times g$-spin for 5 min and homogenized by passing through a 22 G needle in a 1.5× cell pellet volume of B88 buffer (20 mM HEPES-KOH, pH 7.2, 250 mM sorbitol, 150 mM potassium acetate and 5 mM magnesium acetate) plus cocktail protease inhibitors (Roche, Indianapolis, IN), phosphatase inhibitors (Roche) and 0.3 mM DTT. The cell homogenates were centrifuged at $160,000 \times g$ for 30 min, supernatant fractions were collected and the centrifugation was repeated three times to achieve a clarified fraction (approximately 6–10 mg/ml of protein) which was used in the lipidation reaction.

## Purification of HisT7-LC3 and HisT7-LC3 (G/A)

*E. coli* BL21 cells with the indicated expression plasmids were cultured at 37°C overnight. The overnight culture was inoculated at 1:50 dilution to a one liter volume and shaken at 37°C to an OD600 of 0.6–0.8. Protein expression was induced with 100 μM IPTG at 23°C for 5 hr and the cells were collected by centrifuging at $10,000 \times g$ for 10 min. The pellet was washed with 0.1 M PBS and suspended with 20 ml 0.2 M PBS (pH 7.4) with 15 mM imidazole and 1x protease inhibitors (Roche). Lysozyme was added to the cells at a concentration of 0.5 mg/ml and the digestion was performed on ice for 30 min after which Triton X-100 was added to a concentration of 0.5%. The cell suspension was sonicated with five to seven 15 s bursts until the solution was not viscous and the lysate was centrifuged at $100,000 \times g$ for 30 min. The supernatant was collected and incubated with 1 ml Ni Sepharose (packed beads) at 4°C for 2 hr. Then the beads were collected and washed with 70 vol of cold 0.2 M PBS with 25 mM imidazole and 0.2% Tween-20 followed by 10 vol of 0.2 M PBS with 25 mM imidazole. The bound proteins were eluted with 0.2 M PBS with 250 mM imidazole, buffer exchanged to 0.1 M PBS (pH 7.4) and stored at −80°C. Thrombin digestion was performed in the presence of 1 U/ml of thrombin (Roche) at room temperature for 1 hr followed by adding 1 mg/ml AEBSF (Santa Cruz) to deactivate thrombin.

## In vitro lipidation assay

For each reaction, cytosol (2 mg/ml final concentration), ATP regeneration system (40 mM creatine phosphate, 0.2 mg/ml creatine phosphokinase, and 1 mM ATP), GTP (0.15 mM) (*Kim et al., 2005*), 0.2 μg HisT7-LC3 (1-120) or T7-LC3 (1-120) generated by thrombin digestion and different membrane fractions (0.2 mg/ml PC content final concentration) were incubated in a final volume of 30 μl. The reactions were performed at 30°C for the indicated times. Where indicated, compounds or proteins were added to the reactions.

## Membrane fractionation

Cells (ten 15-cm dishes) were cultured to confluence, harvested and homogenized in a 2.7× cell pellet volume of buffer containing 20 mM HEPES-KOH, pH 7.2, 400 mM sucrose and 1 mM EDTA by passing

through a 22 G needle until ~85% lysis analyzed by Trypan Blue staining. Homogenates were either centrifuged at 100,000×*g* for 45 min to collect total membranes or subjected to sequential differential centrifugation at 1,000×*g* (10 min), 3,000×*g* (10 min), 25,000×*g* (20 min) and 100,000×*g* (30 min, TLA100.3 rotor, Beckman) to collect the membranes sedimented at each speed. The PC content of each fraction was measured as described before (*Ge et al., 2011*). Membrane fractions containing equal amounts of PC were used to test LC3 lipidation activity. The 25,000×*g* membrane pellet, which contained the highest activity, was suspended in 0.75 ml 1.25 M sucrose buffer and overlayed with 0.5 ml 1.1 M and 0.5 ml 0.25 M sucrose buffer (Golgi isolation kit; Sigma). Centrifugation was performed at 120,000×*g* for 2 hr (TLS 55 rotor, Beckman), after which two fractions, one at the interface between 0.25 M and 1.1 M sucrose (L fraction) and the pellet on the bottom (P fraction), were separated. Activities of the two fractions were then tested as described above, and the L fraction was selected and suspended in 1 ml 19% OptiPrep for a step gradient containing 0.5 ml 22.5%, 1 ml 19% (sample), 0.9 ml 16%, 0.9 ml 12%, 1 ml 8%, 0.5 ml 5% and 0.2 ml 0% OptiPrep each. Each density of OptiPrep was prepared by diluting 50% OptiPrep (20 mM Tricine-KOH, pH 7.4, 42 mM sucrose and 1 mM EDTA) with a buffer containing 20 mM Tricine-KOH, pH 7.4, 250 mM sucrose and 1 mM EDTA. The OptiPrep gradient was centrifuged at 150,000×*g* for 3 hr (SW 55 Ti rotor, Beckman) and subsequently ten fractions, 0.5 ml each, were collected from the top. Fractions were diluted with B88 buffer and membranes were collected by centrifugation at 100,000×*g* for 1 hr. The activity of each fraction was determined and the distribution of the activity was compared with that of each membrane marker. Membranes containing an equal amount of PC from each fraction were also measured for PE content using an enzymatic assay (*Hokazono et al., 2011*).

## Immunoisolation

Cells (ten 15-cm dishes) expressing indicated protein markers were cultured to confluence and harvested as indicated in the 'Membrane fractionation' section. Membranes from either the 25,000×*g* membrane pellet (for Flag-GFP-ER-TM or LAMP1-RFP-Flag immunoisolation) or the L fraction (for SEC22B, KDELR-Flag or Vangl2-myc immunoisolation) were collected, suspended in immunoisolation buffer containing 25 mM HEPES, pH 7.4, 140 mM potassium chloride, 5 mM sodium chloride, 2.5 mM magnesium acetate, 50 mM sucrose and 2 mM EGTA (*Zoncu et al., 2011*), and diluted to a PC content of 0.2 mg/ml. Anti-Flag (100 µl, packed volume) or anti-Myc agarose (Sigma) was added to a 1 ml membrane suspension with or without 0.2 mg/ml blocking peptides (3xFlag peptide and Myc peptide; Sigma) and mixed by rotation at 4°C overnight. For immunoisolation of endogenous SEC22B vesicles, 20 µl rabbit anti-SEC22B antibody was added to a 1 ml L fraction membrane suspension with or without 0.2 mg/ml SEC22B antibody blocking peptide and incubated for 3 hr at 4°C followed by addition of 100 µl (packed volume) Protein A Sepharose for overnight incubation at 4°C. Beads with the associated membranes were washed with 1 ml immunoisolation buffer three times and membranes bound to the beads were eluted by incubating with 0.5 mg/ml of the indicated competing peptides for 0.5 hr at room temperature. The eluted membranes were collected by ultracentrifugation. The sedimented activities were determined and compared to input membrane of equal PC content.

## Membrane recruitment assay

For cytosol preparation, HEK293T cells were transfected with plasmids encoding the genes for the indicated proteins by X-tremeGene HP (Roche). At 48 hr post-transfection, the cytosols were harvested as described above.

For membrane preparation, *Atg5* KO MEF cells were treated with indicated compounds and the cells were lysed. After a 1,000×*g* centrifugation, the total membranes from the supernatant were sedimented at 100,000×*g* for 1 hr. Similar reactions containing the cytosols, ATP regeneration system, GTP and the total membranes from different treatments were carried out in a final volume of 50 µl at 30°C for 1 hr.

After the reaction, a membrane flotation experiment was performed. OptiPrep (200 µl of 50%) diluted in B88 was added to the reaction mixture to make a 40% solution which was overlayed with 200 µl 35% OptiPrep and 50 µl B88. The gradient was centrifuged at 150,000×*g* for 1.5 hr. Seven fractions, 80 µl each, were collected from the top. The bottom fractions no. 5 to 7 were combined and evaluated by SDS-PAGE and immunoblot to examine the distribution of indicated protein markers.

## Immunofluorescence microscopy, immunoblot and dot blot

Immunofluorescence was performed as previously described (*Ge et al., 2008*, *2011*). Images were acquired with a Zeiss LSM 710 laser confocal scanning microscope. ERGIC recovery quantification was

described previously (*Puri and Linstedt, 2003*). Golgi recovery was quantified by manually counting the percent of cells displaying a perinuclear location of GM130. For each sample, more than 100 cells were counted. Immunoblot was performed as previously described (*Ge et al., 2008*, *2011*) and dot blot was carried out according to the PIP Strip user manual (Echelon). Images were acquired and bands were quantified with Chemidoc MP Imaging System (Bio-Rad).

## Cell culture

HEK293T cells were grown in a tissue culture facility. *Atg5* KO and control MEFs (*Kuma et al., 2004*), *Atg3* KO, *Atg7* KO and control MEFs (*Komatsu et al., 2005*; *Sou et al., 2008*), and *Ulk1* KO and control MEFs (*Kundu et al., 2008*) were generously provided by Noboru Mizushima (Tokyo Medical and Dental University, Japan), Masaaki Komatsu (Tokyo Metropolitan Institute of Medical Science, Japan) and Kundu Mondira (St. Jude Children's Research Hospital). The cells were grown in monolayer at 37°C in 5% $CO_2$ and maintained in Dulbecco's modified Eagle's medium (DMEM) supplemented with 10% FBS. For starvation, the cells were incubated in EBSS for the indicated times in the absence or presence of the drugs indicated in the manuscript.

## Acknowledgements

We thank Qing Zhong, Nicholas Ktistakis, Noboru Mizushima, Masaaki Komatsu, David King, Kundu Mondira, Eisaku Hokazono, and Jennifer Lippincott-Schwartz (NICHD) for reagents; Bob Lesch, Ann Fisher, Xiaozhu Zhang and Amita Gorur (University of California, Berkeley) for technical assistance; Qing Zhong, Jeremy Thorner (University of California, Berkeley), Ta-Yuan Chang (Dartmouth Medical School), David Sabatini, Roberto Zoncu (Whitehead Institute) and Xuejun Jiang (Memorial Sloan-Kettering Cancer Center) for helpful information and advice on the study; and Daniel Klionsky (University of Michigan) and Livy Wilz (University of California, Berkeley) for manuscript editing. LG was supported by a fellowship from the Human Frontier Science Program (HFSP) and the Jane Coffin Childs Fund (JCCF). DM and MZ are HHMI Associates. RS is an Investigator of the HHMI and a Senior Fellow of the University of California, Berkeley Miller Institute.

## Additional information

### Competing interests

RS: Editor-in-Chief, *eLife*. The other authors declare that no competing interests exist.

### Funding

| Funder | Grant reference number | Author |
|---|---|---|
| Howard Hughes Medical Institute | | Randy Schekman |
| University of California, Berkeley Miller Institute | | Randy Schekman |
| Human Frontier Science Program | LT000003/2012 | Liang Ge |
| Jane Coffin Childs Fund | | Liang Ge |

The funders had no role in study design, data collection and interpretation, or the decision to submit the work for publication.

### Author contributions

LG, Conception and design, Acquisition of data, Analysis and interpretation of data, Drafting or revising the article, Contributed unpublished essential data or reagents; DM, Acquisition of data, Drafting or revising the article; MZ, Acquisition of data, Contributed unpublished essential data or reagents; RS, Conception and design, Analysis and interpretation of data, Drafting or revising the article

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
