## [Decision Letter]

Thank you for sending your work entitled “Lipidation of LC3 identifies the ER-Golgi intermediate compartment as a precursor of the mammalian autophagosome” for consideration at *eLife*. Your article has been favorably evaluated by a Senior editor, a Reviewing editor, and two reviewers.

The Reviewing editor and the two reviewers discussed their comments before we reached this decision, and the Reviewing editor has assembled the following comments to help you prepare a revised submission.

We agree that your manuscript represents an elegantly designed and carefully performed study that provides an important new tool to the field (i.e., the development of an in vitro assay system to analyze LC3 lipidation using components of mammalian cells) and increased insight into the mechanism of autophagosome biogenesis (i.e., further clarification of the role of the ERGIC as a membrane determinant of the site of LC3 lipidation).

The reviewers have suggested some studies that would either (a) further increase the impact of your findings or (b) provide additional controls to further strengthen the findings. Here, we provide a summary of the suggestions. In our collective view, the most crucial point is to provide additional data such as EM analysis (or another approach) to determine whether your in vitro system for studying LC3 lipidation results in actual autophagosome reconstitution.

Suggestions to increase the impact/significance of the findings:

1) As mentioned above, further characterization of the cell-free system would be important. Specifically, does the reconstitution system allow for the formation of mature autophagosomes and autolysosomes? Protease protection assays, EM analysis, and analysis of the effects of lysosomal inhibitors should be considered to strengthen this aspect of the study.

2) EM analysis of the tissue culture experiments would help determine whether the ERGIC compartment directly contributes to autophagosome formation.

Suggestions for additional controls to further strengthen your data:

1) The lipidation of LC3 requires the Atg12–5–16 complex. Using gel filtration of the cytosol, we suggest that you demonstrate that this complex is formed or maintained in your reconstitution systems using 1) ERGIC membrane fractions, and 2) a membrane fraction you have ruled out as being a membrane source (see Figure 8), such as ER, late endosomes and plasma membrane. This would be an important control to show the integrity of the E3 activity in the cytosol even though the lipidation is reduced, and eliminate the possibility that the membrane fractions or assay conditions can deplete or contribute inhibitory factors, which may be altering the lipidation activity of the Atg12–5–16 complex.

2) In light of the recent data about the MAM and autophagosome biogenesis (20), you should consider validating your MAM fractionation for Atg14L by Western blot.

3) One of the most important experiments is the recruitment of Atg14 and DFCP1 to the ERGIC in Figure 12. These data are not as robust as the rest of the manuscript and it would be important to show that a transfected negative control (e.g., GFP alone) does not float up on the ERGIC membranes. In addition, you might consider testing GFP–LC3–II recruitment as an additional further validation in this assay.

[Editors’ note: before acceptance, the following revisions were also requested.]

There is one remaining issue that needs to be addressed before acceptance, as outlined below:

We agree that the formation of a closed concentric membrane sphere (autophagosome) is likely to be quite complex, and we are willing to accept your argument that the in vitro reconstitution of this process is beyond the scope of the present paper. Nonetheless, in order for this paper to have the proper impact in the field and provide as useful a tool as possible for future researchers, we think that it is important to provide as much characterization as you can in the text of what your system is reconstituting in vitro. Accordingly, we would like you to mention (and discuss as you deem appropriate) the results of LC3-II protease protection experiments in the text. As you point out, the absence of complete reconstitution of a closed membrane sphere does not detract from the overall importance of your work; however, we feel that it is important for readers to know this limitation of your in vitro LC3 lipidation system.

---

## [Author Response]

[Suggestions to increase the impact/significance of the findings:]

*1) As mentioned above, further characterization of the cell-free system would be important. Specifically, does the reconstitution system allow for the formation of mature autophagosomes and autolysosomes? Protease protection assays, EM analysis, and analysis of the effects of lysosomal inhibitors should be considered to strengthen this aspect of the study*.

We too would like to have our cell-free reaction recapitulate the entire pathway of authophagosome biogenesis, but this goal may be a bit much for a first report. Nonetheless, we have performed a protease protection assay hoping to observe the progressive enclosure of lipidated LC3 within a sealed compartment as might be expected when the autophagosome precursor membrane closes to segregate cytosolic components. Unfortunately, on repeated tests in prolonged incubations, we find little evidence for reproducible protease protection of the LC3-II species. We are not discouraged by this because the overall pathway is likely to be quite complex, including the fusion of additional membranes carrying such proteins as ATG9 (which is missing from our enriched ERGIC membrane), as well as a substantial morphological transformation from a membrane sheet into a closed concentric membrane sphere. An alternative strategy could be to start with membranes from an *Atg* null cell line blocked at a distinctly later step in the pathway. We hope the reviewers will appreciate that this is beyond the scope of this paper, which we believe is already quite complex.

*2) EM analysis of the tissue culture experiments would help determine whether the ERGIC compartment directly contributes to autophagosome formation*.

We thank the reviewers for the suggestion. Our antibodies are not good enough for immunogold labeling in EM analysis. Instead, we performed confocal microscopy to analyze the colocalization of ERGIC with ATG14 and DFCP1, two early autophagy markers as indicated in the manuscript. We found that the starvation-induced puncta of ATG14 partially overlaps with ERGIC (Figure 12—figure supplement 2). DFCP1, the autophagy-related PI3P binding protein, colocalizes with ERGIC shortly after starvation (Figure 12—figure supplement 2). These data are consistent with our membrane recruitment assay in Figure 12. Together, the results suggest that ATG14 may transiently associate with ERGIC to generate PI3P, and subsequently be recognized by DFCP1 to initiate autophagosome biogenesis.

[Suggestions for additional controls to further strengthen your data:]

*1) The lipidation of LC3 requires the Atg12-5-16 complex. Using gel filtration of the cytosol, we suggest that you demonstrate that this complex is formed or maintained in your reconstitution systems using 1) ERGIC membrane fractions, and 2) a membrane fraction you have ruled out as being a membrane source (see Figure 8), such as ER, late endosomes and plasma membrane. This would be an important control to show the integrity of the E3 activity in the cytosol even though the lipidation is reduced, and eliminate the possibility that the membrane fractions or assay conditions can deplete or contribute inhibitory factors, which may be altering the lipidation activity of the Atg12–5–16 complex*.

We thank the reviewers for the suggestion. We did the lipidation assay with ERGIC membrane fractions and the P fraction enriched in ER. After the reaction, we sedimented the membranes and collected the supernatant fractions followed by chromatography on a Superpose 6 column. Fractions were collected and immunoblots were performed to examine the distribution of ATG5–12 and ATG16. We found that the ATG5–12–16 complex is formed and maintained in both conditions (Figure 7—figure supplement 1). Therefore, the distinct lipidation activity observed with different membrane fractions is not due to an alteration of ATG5–12–16 complex.

*2) In light of the recent data about the MAM and autophagosome biogenesis (20), you should consider validating your MAM fractionation for Atg14L by western blot*.

We thank the reviewers for the suggestion. We performed an immunoblot with ATG14 antibody and detected a tiny amount of ATG14 in the MAM fraction under normal conditions (Figure 8—figure supplement 1). However, after starvation, we observed an increase of ATG14 in the MAM faction (Figure 8—figure supplement 1). The observation is consistent with [20].

*3) One of the most important experiments is the recruitment of Atg14 and DFCP1 to the ERGIC in Figure 12. These data are not as robust as the rest of the manuscript and it would be important to show that a transfected negative control (e.g., GFP alone) does not float up on the ERGIC membranes. In addition, you might consider testing GFP–LC3–II recruitment as an addition further validation in this assay*.

We thank the reviewers for the suggestion. In Figure 12—figure supplement 1, we validated the assay by showing that the recruitment of ATG14 and DFCP1 is regulated similarly to cellular autophagy, including starvation stimulation and PI3K dependency. We also showed that tubulin, an abundant cytosolic protein, was barely recruited to the membrane. For the LC3-II recruitment, we took the reviewer’s suggestion and added the T7-tagged LC3-I in the in vitro recruitment reaction. We found that the lipidated T7-LC3 was recruited to the membrane fraction while the soluble form LC3-I was not. Moreover, depletion of ERGIC mitigated the generation of LC3-II as indicated by the reduced signal in the top membrane fraction (Figure 12).

*[Editors’ note: before acceptance, the following revisions were also requested.*]

*There is one remaining issue that needs to be addressed before acceptance, as outlined below*:

*We agree that the formation of a closed concentric membrane sphere (autophagosome) is likely to be quite complex, and we are willing to accept your argument that the in vitro reconstitution of this process is beyond the scope of the present paper. Nonetheless, in order for this paper to have the proper impact in the field and provide as useful a tool as possible for future researchers, we think that it is important to provide as much characterization as you can in the text of what your system is reconstituting in vitro. Accordingly, we would like you to mention (and discuss as you deem appropriate) the results of LC3-II protease protection experiments in the text. As you point out, the absence of complete reconstitution of a closed membrane sphere does not detract from the overall importance of your work; however, we feel that it is important for readers to know this limitation of your in vitro LC3 lipidation system*.

We thank the reviewers for appreciating the complexity of a mature autophagosome formation in vitro, which is beyond the scope of this paper. We have pointed out the limitation of our system in the Results and Discussion sections.